# UNLEARNING MAPPING ATTACK: EXPOSING HIDDEN VULNERABILITIES IN MACHINE UNLEARNING

## ABSTRACT

As machine learning becomes increasingly data-dependent, concerns over privacy and content regulation among data owners have intensified. Machine Unlearning has emerged as a promising solution, allowing for the removal of specific data from pre-trained systems to protect user privacy and regulate information. Existing research on Machine Unlearning has shown considerable success in eliminating the influence of certain data while preserving model performance. However, the resilience of Machine Unlearning to malicious attacks has not been thoroughly examined. In this paper, we investigate the hidden vulnerabilities within current Machine Unlearning techniques. We propose a novel adversarial attack, the Unlearning Mapping Attack (UMA), capable of undermining the unlearning process without altering its procedures. Through experiments on both generative and discriminative tasks, we demonstrate the susceptibility of existing unlearning techniques to UMA. These findings highlight the need to reassess unlearning objectives across various tasks, prompting the introduction of a Robust Unlearning standard that prioritizes protection against adversarial threats. Our extensive studies show the successful adaptation of current unlearning methods to this robust framework. The Python implementation will be made publicly available.

## 1 INTRODUCTION

In recent years, the rapid growth of Machine Learning (ML) and Deep Neural Networks (DNNs) has led to significant advancements in various fields, including image classification (Krizhevsky et al., 2012; Sahoo et al., 2019), natural language processing (Xu et al., 2019; Radford et al., 2018), and autonomous driving (Litman, 2020). As DNNs become increasingly data-dependent, personal data may be involved in model training, raising critical privacy concerns. In response to privacy regulations such as the General Data Protection Regulation (GDPR) (Regulation, 2018) and the California Consumer Privacy Act (CCPA) (Pardau, 2018), the concept of Machine Unlearning (MUL) has emerged as a potential solution. MUL aims to remove specific data from trained models, allowing for the "right to be forgotten." Beyond privacy concerns, content regulation has become another key motivation for machine unlearning (Kurmanji et al., 2024; Shumailov et al., 2024). To remove impermissible knowledge such as unlicensed copyrighted material (Yao et al., 2023), malicious information (Yao et al., 2023), or harmful capabilities (Shumailov et al., 2024) from models, machine unlearning ensures that the unlearned models align with ethical and legal standards.

While existing MUL techniques have demonstrated strong performance in eliminating the influence of specific data on both privacy-sensitive and content-sensitive tasks (Warnecke et al., 2021; Li et al., 2024; Graves et al., 2021; Tarun et al., 2023; Golatkar et al., 2020; Liu et al., 2022), they often overlook a critical vulnerability: the susceptibility of unlearned models to malicious attacks. This gap raises a crucial question regarding the robustness of current MUL methods:

*Are unlearned systems sufficiently resilient to withstand malicious attempts aimed at recovering or exploiting the unlearned information?*

Recent research has explored this question by focusing on specific domains such as diffusion models (DMs) and large language models (LLMs). In the context of DMs, several approaches have been proposed to exploit vulnerabilities in content erasure (Zhang et al., 2025; Tsai et al., 2024; Han et al., 2024; Pham et al., 2023), yet none address potential remedies for these threats. The most

relevant to ours is the concurrent study (Yuan et al., 2024). It leverages adversarial Suffix prompting to compromise unlearned LLMs, but does not extend its analysis beyond LLMs.

To address this critical question in a broader context, particularly including various discriminative tasks and generative tasks in the computer vision domain, we revisit the concept of machine unlearning and introduce an unified framework to investigate the hidden weaknesses of current MUL methods: successfully unlearned knowledge can resurface through appropriate probes. To address this limitation, we introduce and formulate the concept of robust unlearning and a novel post-MUL adversarial attack, namely Unlearning Mapping Attack (UMA), for empirical verification of MUL. Unlike traditional attempts to alter the MUL process or to tamper with the data to be unlearned, our UMA attack targets failing unlearned models (i.e. reintroducing the unlearned knowledge) without being involved in the unlearning process. As a post-MUL attack, UMA only requires access to the model before and after the unlearning process, which practically makes the model provider or the MUL service provider an ideal candidate for carrying out such an attack. Our experiments, conducted on both discriminative and generative tasks using datasets such as CIFAR-10, CIFAR-100, Tiny-ImageNet, and ImageNet-1k, demonstrate that even when a model appears to have forgotten specific data, UMA can successfully retrieve the forgotten information, revealing a critical vulnerability in existing unlearning techniques.Furthermore, to mitigate this threat, we performed preliminary studies to incorporate the robust unlearning concept into current unlearning methods to prevent adversaries from recovering forgotten data, ensuring that unlearned models remain secure even in the face of advanced attacks like UMA.

Our contribution in this study can be summarized as follows:

- We revisit the definition of Machine Unlearning, and formulate the novel perspective: robust unlearning.
- We introduce a post-MUL malicious attack, UMA, to undermine Machine Unlearning, exposing a critical threat in existing unlearning methods. It compromises the unlearned systems without requiring any modifications to the data or the unlearning process, thus positioning it as a superior evaluation metric for machine unlearning verification.
- We present extensive empirical efforts to mitigate the threat on the basis of current MUL, enhancing system robustness against such vulnerability.

## 2 RELATED WORK

**Machine Unlearning.** Machine unlearning (Cao & Yang, 2015) was initially introduced as a method to remove the influence of specific data points from machine learning models, driven by privacy and security concerns. The most straightforward unlearning method involves retraining the model from scratch without the undesired data. However, as models like LLMs increase in size and complexity, this retraining approach has become computationally prohibitive. To address this, researchers have developed alternative methods, including exact unlearning and approximate unlearning. Exact unlearning (Golatkar et al., 2020) aims to produce a model that behaves indistinguishably from a model trained from scratch without the forgotten data. For instance, SISA (Bourtoule et al., 2021) proposes to train separate models using multiple disjoint data shards and achieves exact learning by retraining the corresponding shard models associated with unlearning requests. Considering scalability and cost-efficiency, approximate unlearning provides a more practical alternative, relaxing some constraints to improve efficiency while maintaining acceptable performance. For example, the first-order unlearning leverages the first-order Taylor series expansion of model parameters to compute gradient updates for MUL, while the second-order method incorporates the inverse Hessian matrix for more accurate parameter adjustments (Warnecke et al., 2021). To enhance stability and accuracy across different domains while maintaining computational efficiency, recently, SalUn (Fan et al., 2024) employed a weight saliency map, assigning different importance levels to model parameters, allowing them to be updated at varying rates. Despite these advancements, existing methods remain vulnerable to sophisticated attacks aimed at extracting forgotten information, leading to significant privacy and security concerns.

**Attacks to MUL.** The unlearning target usually deals with privacy-related or security-sensitive data, it is naturally assumed to be at risk of malicious attacks by adversaries. In fact, prior research has already explored various malicious attempts targeting MUL. Adversarial text prompts and Concept

Inversion attacks are particularly investigated to undermine DMs for content erasure (Zhang et al., 2025; Tsai et al., 2024; Han et al., 2024; Pham et al., 2023). Targeting more generic unlearning scenarios, Qian et al. (2023) examines the possibility of injecting small, targeted noise into forget samples within the forget set. This manipulation leads the unlearned model to fail in classification tasks. Liu et al. (2024a) induces backdoor behavior in a model through the standard MUL process with selected data. Thudi et al. (2022); Zhang et al. (2024) examine how attackers can deceive others into believing the MUL process has been completed when, in reality, the model has not forgotten the specified samples, or the unlearning process was never carried out. They leverage techniques from Data Ordering Attacks(Shumailov et al., 2021) to falsify Proof-of-Unlearning (PoUL), with the intent of either enhancing model performance or reducing computational costs. Different from these works either editing the unlearning data or modifying the unlearning algorithms, this study investigates post-MUL attacks to expose the hidden vulnerabilities in existing machine unlearning approaches.

**Unlearning Verification.** To evaluate the efficacy of an unlearning algorithm, various verification methods are proposed. *Attack-based verification* simulates a malicious attacker attempting to resurface or extract sensitive information from the unlearned model. For instance, membership inference attacks (MIA) (Shokri et al., 2017) are explored to determine whether a specific sample was part of the training data. Similarly, model inversion attacks (Fredrikson et al., 2015) try to retrieve input samples given the model's outputs, potentially leaking class information to the attacker. Backdoor attacks, or backdoor verifications, are also widely used to assess unlearning efficacy (Sommer et al., 2020; Gao et al., 2024; Guo et al., 2024). Alternatively, following Proof of Learning (PoL) (Jia et al., 2021), *reproducing verification* (Zhang et al., 2024) works by recording logs during the unlearning operation, allowing users or third-party inspectors to reproduce the process and verify its authenticity, ensuring that the unlearning was executed as claimed. Beyond the above two categorical verification approaches, accuracy verification (Golatkar et al., 2020; Mehta et al., 2022) quantifies the unlearning performance on the retaining and forgetting data, and Relearning time verification (Tarun et al., 2023) infers the unlearning efficacy by counting the elapsed time to relearning the forgotten information on the unlearned models. In this study, we introduce the UMA attack for evaluating unlearning efficacy.

## 3 ROBUST MACHINE UNLEARNING: PRELIMINARIES

Before introducing the concept and formulation of robust machine unlearning, let's first revisit the standard definition of machine unlearning.

### 3.1 VULNERABILITY IN CONVENTIONAL MACHINE UNLEARNING

**Notation.** Let $\mathcal{D} = \{(x_i, y_i)_{i=1}^N\}$ denote the training dataset used to train a model $f(\cdot; \theta)$, where $N$ represents the number of training samples, each consisting of input features $x \in \mathbb{R}^d$ and target output $y$, and $\theta$ denotes the model's parameters. The model's training process is represented as $\mathcal{A}(f(\cdot; \theta), \mathcal{D})$, while the unlearning process is denoted by $\mathcal{U}(f(\cdot; \theta), \mathcal{D}_u)$, where $\mathcal{D}_u \subset \mathcal{D}$ is the subset of data to be unlearned. After the unlearning process, the updated model is expressed as $f_u(\cdot; \theta^u)$, i.e., $f_u(\cdot; \theta^u) = \mathcal{U}(f(\cdot; \theta), \mathcal{D}_u)$.

**Definition 1** *(Machine Unlearning (Cao & Yang, 2015).) An unlearning process $\mathcal{U}(f(\cdot; \theta), \mathcal{D}_u)$ aims to find an unlearned model $f_u(\cdot; \theta^u)$ so that it closely aligns with a model trained from scratch on the retain set $\mathcal{D}_r = \mathcal{D}/\mathcal{D}_u$, i.e. $f_u(\cdot; \theta^u) = f_r(\cdot; \theta^r) = \mathcal{A}(f(\cdot; \theta), \mathcal{D}_r)$.*

Based on Definition 1, an unlearning algorithm should align closely with a retrained model on the retain set. However, for large-scale systems like foundation models whose retraining is computationally prohibitive or impractical for information unlearning, a commonly used, practical strategy for effective forgetting is to diverge an unlearned model sufficiently from its prior behavior on the forget set while preserving its performance on the retain set. For instance, a generative unlearned model should be incapable of generating undesirable information in the forget set (Warnecke et al., 2021; Fan et al., 2024; Li et al., 2024). In other words, empirically, an unlearned model should decorrelate the input from the original output by a significant margin, $\varepsilon_1$, while maintaining close predictions for $x \in \mathcal{D}_r$. Specifically,

$$E_{x \in \mathcal{D}_u}[||f_u(x, \theta^u) - f(x, \theta)||] > \varepsilon_1, E_{x \in \mathcal{D}_r}[||f_u(x, \theta^u) - f(x, \theta)||] \leq \varepsilon_2, \tag{1}$$

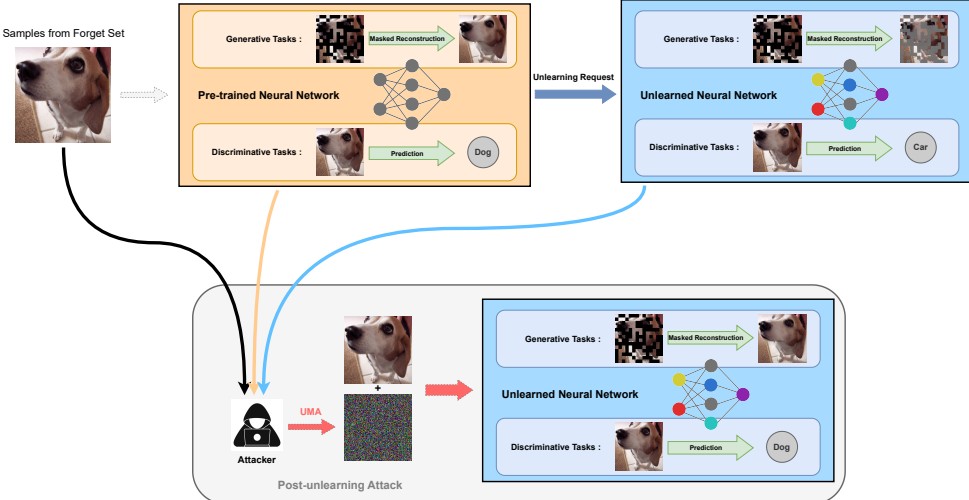

Figure 1: An illustration of the novel malicious post-MUL attack. With knowledge of the pre- and post-unlearning models, the attacker attempts to recover forgotten information from the unlearned neural network by injecting carefully-designed noise into the query input.

where $E$ represents the statistical mean over a distribution, and $\varepsilon_1$ and $\varepsilon_2$ may vary depending on the task and should align with real-world attack detectability and interpretability of human visual inspection. However, Definition 1 and the conventional conditions formulated in (1) have significant safety concerns, particularly regarding whether an unlearned model might reproduce the unlearned information when provided with any input.

Let's first consider the case of a generative model, which produces outputs based on various prompts or inputs. In the context of generative tasks, the primary objective of unlearning is to prevent the model from generating unlearned knowledge Warnecke et al. (2021); Fan et al. (2024); Li et al. (2024). However, the standard definition of machine unlearning lacks constraints that guarantee the unlearned model cannot resurface the unlearned knowledge when exposed to arbitrary or carefully designed input prompts (Shumailov et al., 2024). This safety concern, formalized in Proposition 2, is critical in the context of using machine unlearning for knowledge regulation. We demonstrate this safety threat in Fig. 1.

**Proposition 2** *For an unlearned generative system $f_u(\cdot, \theta^u)$ that satisfies conditions in (1), there may exist an input $\delta_x \notin \mathcal{D}_u$ s.t. $||f_u(\delta_x, \theta^u) - f(x, \theta)|| < \varepsilon_1, \forall x \in \mathcal{D}_u$.*

This safety risk also extends to machine unlearning in discriminative models, with slight modifications outlined in Proposition 3. Discriminative tasks, which focus on recognizing objects in a given query image, have the goal of preventing the model from identifying specific inputs post-unlearning. Naturally, this assumes that the inputs to the original model $f(\cdot, \theta)$ and the unlearned model $f_u(\cdot, \theta^u)$ share significant semantic features. However, if slight perturbations $\delta_x$ on the data $x$ can bypass the unlearning process, the system's security is compromised.

**Proposition 3** *For an unlearned discriminative model $f_u(\cdot, \theta^u)$ that satisfies conditions in (1), there may exist a small non-zero $\delta_x \in \mathbb{R}^d$ (i.e. $||\delta_x|| < \epsilon$) s.t. $||f_u(x + \delta_x, \theta^u) - f(x, \theta)|| < \varepsilon_1, \forall x \in \mathcal{D}_u$.*

Note, $\delta_x$ in both Propositions are data-specific and model-specific. The extra constraint $||\delta_x|| < \epsilon$ for discriminative models ensures semantic similarity between $x$ and $x + \delta_x$, so that the attack remains meaningful and realistic, aligning with its practical use of these models. By contrast, generative models allow for unconstrained $\delta_x$, as the attack focuses exclusively on the outputs generated from crafted inputs, irrespective of input realism.

In sum, Propositions 2 and 3 emphasize that even if an unlearned system successfully decorrelates from the original output for a given input $x$, attackers could potentially craft inputs that cause the unlearned model to revert to outputs resembling those of the original model.

## 3.2 Robust Unlearning

The limitations and risks discussed in the previous section underscore significant safety concerns with unlearned models, particularly their vulnerability to malicious exploitation for knowledge resurfacing. While current methods generally satisfy the conditions in (1), they fail to guarantee that all traces of the unlearned data are entirely removed from the model, as emphasized in Proposition 2 and 3. Furthermore, given that machine unlearning is inherently tied to security, it is crucial that unlearning algorithms exhibit strong resistance to adversarial attacks. In response, we propose a new definition, Robust Unlearning, that is adaptable across various machine learning contexts and resilient to malicious attacks, offering a more comprehensive and secure approach to unlearning.

**Definition 4** (*Robust Unlearning*). *A unlearning process, $\mathcal{U}(f(\cdot; \theta), \mathcal{D}_u)$, is considered robust if $\forall x \in \mathcal{D}_u$, $\forall \delta_x \in \mathbb{R}^d$, we have conditions in (1), plus $||f_u(\delta_x, \theta^u) - f(x, \theta)|| > \varepsilon_1$ for generative tasks and $||f_u(x + \delta_x, \theta^u) - f(x, \theta)|| > \varepsilon_1$ for discrimnative models (where $||\delta_x|| < \epsilon$).*

Intuitively, Robust Unlearning ensures that the system is incapable of producing the specified information, whether under normal conditions or in the presence of adversarial manipulation. We propose this as a comprehensive and robust standard for defining and evaluating unlearning algorithms. It is important to note, however, that the commonly used retraining method, a baseline in many existing unlearning techniques, does not inherently satisfy the criteria for Robust Unlearning. Due to the generalization capabilities of neural networks, even models trained without specific data may still generate corresponding information Shumailov et al. (2024), thereby compromising the effectiveness of the unlearning process. To identify MUL-specific vulnerabilities, the unlearned model can be compared to a retrained model using the Robust Unlearning criteria in Definition 4. For instance, our empirical experiments on discriminative models with class-wise unlearning in Table 1 show that retraining achieves strong robustness, while those evaluated unlearning algorithms are less robust to adversarial attacks, suggesting these vulnerabilities are intrinsic to the unlearning methods.

## 4 Unlearning Mapping Attack

Building on this foundation of Robust Unlearning in the previous section, this section formally introduces our Unlearning Mapping Attack. It can serve two primary purposes in the context of machine learning. First, UMA generates malicious perturbations tailored to unlearned models, resurfacing forgotten knowledge for adversarial use. Second, UMA is a practical and actionable verification tool for benchmarking unlearning methods. By applying UMA to various unlearning algorithms with the same set of forget data, we can quantitatively assess their vulnerabilities against adversarial prob.

### 4.1 Threat Model

Generally, in a machine unlearning workflow, the individuals involved can be divided into two groups: servers and participants (Liu et al., 2024b). Servers include model and service providers, who are responsible for providing the model or inference platform, while participants consist of data contributors and general users. While much of the existing research has focused on threats from participants, such as information leakage (Chourasia & Shah, 2023; Chen et al., 2021), adversarial unlearning (Liu et al., 2024a; Qian et al., 2023), and sabotage the unlearning process (Marchant et al., 2022), only a limited number of studies have examined unlearning threats from the server side(Thudi et al., 2022; Zhang et al., 2024).

In the unlearning threat we formulate in the previous section, the attacker is considered to be in the role of the service provider. We assume the attacker has full knowledge of the model *before and after* the unlearning process, as well as access to specific samples that need to be unlearned. Unlike previous studies where the service provider may attempt to fake unlearning to improve model performance or lower computational cost Zhang et al. (2024), here, we assume the attacker's goal is to extract the forgotten information from the unlearned models. Given the increasing complexity of deep neural network architectures such as LLMs, it becomes nearly impossible to trace the exact amount of knowledge that remains within the network after unlearning, making these models highly susceptible to such attacks. Note, while the server is the most likely adversary due to its privileged access to the white-box attack setting, the threat is not strictly limited to the server. Anyone with access to the knowledge could theoretically execute UMA.

## 4.2 ATTACK FORMULATION

Recall that reproducing verification, such as Proof of Unlearning (PoUL), is a unique form of unlearning verification designed to ensure transparency from the model training side by requiring the model trainer to report model parameters, data used, and the unlearning algorithm applied. While Thudi et al. (2022) points out that a forged map can be easily created to fake PoUL, the potential for malicious unlearning attempts from the server side is still limited, if not completely prevented.

Here, we propose an unlearning mapping attack, which focuses on uncovering residual information within the neural network rather than injecting new information. This attack method enables us to map the original output to a new input, allowing the unlearned neural network to still produce information that should have been removed. In this context, the UMA attack can be formulated as

$$\arg\min_{\delta_x} ||f_u(\delta_x, \theta^u) - f(x, \theta)||, \forall x \in \mathcal{D}_u. \tag{2}$$

Note that this attack method only modifies the perturbation $\delta_x$ to achieve malicious information mapping. It does not require any change to the unlearning algorithm or the model parameters after the unlearning process. As a result, it would not be restricted by PoUL or any other existing verification method, making it difficult to detect and counter.

The UMA formulation in (2) aligns with the definition of robust unlearning. If for every $x \in \mathcal{D}_u$ we find an optimal $\delta_x$ to minimize the difference, and the minimum difference is still larger than $\varepsilon_1$, we can conclude that the unlearned model is robust with respect to $\varepsilon_1$. This consistency indicates that, under ideal conditions (e.g., if the optimization objective is convex), UMA provides a theoretical guarantee of robustness.

## 4.3 GRADIENT-BASED IMPLEMENTATION OF UMA

To achieve the attack goal outlined in (2), we introduce a gradient-based input mapping attack:

$$\arg\min_{\{\delta_x\}} E_{x \in \mathcal{D}_u} \left[ \mathcal{L}(f_u(\delta_x; \theta^u), f(x; \theta)) \right], \tag{3}$$

where $\mathcal{L}$ quantifies the difference and can vary depending on the context, such as Mean Square Error Loss, Binary Cross-Entropy Loss, or KL Divergence Loss. To solve this optimization problem, we adopt the Projected Gradient Descent (PGD) method proposed in Madry et al. (2017) as a baseline to find the input $\delta$. While PGD is normally used to maximize empirical loss, in this case, we aim to minimize the loss, thus taking the opposite direction of the gradient update:

$$\delta_x^{t+1} = \delta_x^t - \alpha \cdot sign[\nabla_{\delta_x} L(f_u(\delta_x^t; \theta^u), f(x; \theta)], \tag{4}$$

where $\alpha$ stands for the step size for each iteration. The pseudocode of the detailed input mapping attack process is provided in Algorithm 1. For simplicity, we only adopt the PGD-based mapping method as our baseline, though other optimization techniques can be substituted for potentially better performance.

---

**Algorithm 1** Unlearning Mapping Attack

1: **Input:** Pre-trained model $f(\cdot; \theta)$, Unlearned model $f_u(\cdot; \theta^u)$, Unlearning dataset $\mathcal{D}_u$, Attack steps $T$, Attack step size $\eta$
2: **Output:** Attack dataset $\mathcal{D}_{atk}$
3: Random initialize attack noise $\{\delta_x\}$ for $x \in \mathcal{D}_u$
4: **for** $k = 0$ **to** $T$ **do**
5:     Calculate loss $\psi \leftarrow \sum_{x \in \mathcal{D}_u} \mathcal{L}(f(x; \theta), f_u(\delta_x^k; \theta^u))$
6:     Update attack noise $\{\delta_x^{k+1}\} \leftarrow \{\delta_x^k\} - \eta \cdot sign(\nabla_{\delta_x} \psi)$
7:     $\{\delta_x^{k+1}\} \leftarrow clip(\{\delta_x^{k+1}\}, 0, 1)$
8: **end for**
9: Construct attack dataset $\mathcal{D}_{atk} \leftarrow (\delta_x^{k+1}, y_x)$

---

Though UMA and Robust Unlearning are consistent in principle, the optimization problem is typically non-convex. As a result, Algorithm 1 does not guarantee exploration of all possible perturbations. Yet, it still provides a practical and actionable framework for identifying vulnerabilities in unlearning methods. Even in cases where UMA does not succeed, the absence of successful attacks strengthens the empirical evidence that the model may satisfy the robust unlearning criteria.

| CIFAR10 | No Atk | | | $\epsilon = 8/255$ | | $\epsilon = 16/255$ | |
|---|---|---|---|---|---|---|---|
| | TA | UA | MIA | UA | MIA | UA | MIA |
| Original | 94.13 | 100 | 0.9796 | - | - | - | - |
| retrain | 93.99 | 0 | 0 | 0 | 0.0012 | 0 | 0.0020 |
| FT | 92.21 | 22.20 | 0.0928 | 99.96 | 0.9934 | 100 | 0.9892 |
| RL | 91.70 | 0 | 0 | 18.62 | 0.0636 | 56.02 | 0.2534 |
| IU | 89.73 | 21.38 | 0.1522 | 99.68 | 0.9684 | 99.90 | 0.9780 |
| $l_1$-sparse | 91.53 | 0 | 0 | 73.00 | 0.3680 | 97.12 | 0.6886 |
| SalUn | 92.18 | 0 | 0.0002 | 6.04 | 0.0292 | 32.58 | 0.1986 |

| CIFAR100 | No Atk | | | $\epsilon = 8/255$ | | $\epsilon = 16/255$ | |
|---|---|---|---|---|---|---|---|
| | TA | UA | MIA | UA | MIA | UA | MIA |
| Original | 75.25 | 100 | 0.9908 | - | - | - | - |
| retrain | 70.15 | 0 | 0 | 0 | 0 | 0 | 0 |
| FT | 71.38 | 57.28 | 0.5244 | 99.88 | 0.9960 | 99.96 | 0.9968 |
| RL | 71.65 | 3.04 | 0.1976 | 84.84 | 0.5172 | 97.52 | 0.6792 |
| IU | 70.88 | 79.16 | 0.7620 | 99.96 | 0.9852 | 99.92 | 0.9916 |
| $l_1$-sparse | 70.16 | 38.28 | 0.2072 | 99.96 | 0.9400 | 99.96 | 0.9452 |
| SalUn | 72.07 | 2.00 | 0.1976 | 88.24 | 0.6500 | 98.28 | 0.8104 |

| Tiny-ImageNet | No Atk | | | $\epsilon = 8/255$ | | $\epsilon = 16/255$ | |
|---|---|---|---|---|---|---|---|
| | TA | UA | MIA | UA | MIA | UA | MIA |
| Original | 64.17 | 99.96 | 1 | - | - | - | - |
| retrain | 59.41 | 0 | 0 | 0 | 0 | 0 | 0 |
| FT | 58.89 | 22.18 | 0.4192 | 99.94 | 0.9986 | 99.92 | 0.9984 |
| RL | 54.24 | 0.06 | 0.0096 | 90.32 | 0.4230 | 99.42 | 0.6726 |
| IU | 54.86 | 73.94 | 0.7932 | 99.94 | 0.9996 | 99.92 | 0.9998 |
| $l_1$-sparse | 55.76 | 0.46 | 0.0066 | 96.10 | 0.1108 | 99.88 | 0.1306 |
| SalUn | 54.68 | 0 | 0 | 59.86 | 0.2618 | 90.48 | 0.5630 |

Table 1: The Test Accuracy (TA), Unlearning Accuracy (UA), and MIA score of different baselines before and after Unlearning Mapping Attack for the Class Unlearning scenario. The attack is bounded with a maximum strength of 8/255 and 16/255. The original here indicates the model performance before unlearning.

## 5 EXPERIMENTS

In this section, we first evaluate our attack method on existing unlearning methods for both discriminative and generative tasks. Then we conduct a preliminary attempt to improve the robustness of existing machine learning by integrating the robust unlearning concept into the standing MUL process.

### 5.1 EXPERIMENT SETUP

For **discriminative model unlearning**, we choose ResNet50 as our model backbone and conduct experiments on CIFAR10, CIFAR100, and Tiny-ImageNet datasets. We consider both class-wise and instance-wise unlearning in our experiments. For class-wise unlearning, following convention, we select class 0 for CIFAR10, class 5,16,25,34,78 for CIFAR100, and class 4,12,66,72,97,143,149,165,175,190 for Tiny-ImageNet as the unlearning target. For instance-wise unlearning, we randomly select 5000 samples for CIFAR10 and CIFAR100, 10000 samples for Tiny-ImageNet to evaluate the efficacy of data forgetting. As mentioned earlier, the attack for discrimination tasks unlearning needs to be bounded or it would be meaningless otherwise. Therefore, we bound the perturbation to 8/255 and 16/255, which represent the maximum noise intensity while preserving major feature information. In the discrimination unlearning experiment, we adopt several existing unlearning methods as baselines, which include FT (Warnecke et al., 2021), RL (Golatkar et al., 2020), IU (Koh & Liang, 2017; Izzo et al., 2021), l1-sparse (Jia et al., 2023) and SalUn (Fan et al., 2024). We use both Unlearning Accuracy (UA) and Membership Inference Attack (MIA) as the evaluation metrics. Note that the Unlearning Accuracy is considered as the unlearning success rate in some existing works, where they consider UA the higher the better. However, the Unlearning

| CIFAR10 | No Atk | | | 8/255 | | 16/255 | |
|---|---|---|---|---|---|---|---|
| | TA | UA | MIA | UA | MIA | UA | MIA |
| Original | 94.13 | 100 | 0.9732 | - | - | - | - |
| retrain | 93.34 | 93.78 | 0.8636 | 99.98 | 0.9774 | 99.98 | 0.9728 |
| FT | 92.13 | 98.02 | 0.9124 | 99.98 | 0.9794 | 99.96 | 0.9810 |
| RL | 89.22 | 91.88 | 0.8012 | 99.96 | 0.9896 | 100 | 0.9866 |
| IU | 89.82 | 97.92 | 0.8926 | 99.98 | 0.9630 | 99.98 | 0.9628 |
| $l_1$-sparse | 91.32 | 95.76 | 0.8848 | 99.98 | 0.9842 | 100 | 0.9814 |
| SalUn | 90.55 | 93.48 | 0.8140 | 100 | 0.9884 | 99.98 | 0.9872 |

| CIFAR100 | No Atk | | | 8/255 | | 16/255 | |
|---|---|---|---|---|---|---|---|
| | TA | UA | MIA | UA | MIA | UA | MIA |
| Original | 75.25 | 100 | 0.9924 | - | - | - | - |
| retrain | 73.92 | 72.72 | 0.7354 | 99.92 | 0.9910 | 100 | 0.9932 |
| FT | 70.90 | 96.44 | 0.9436 | 99.96 | 0.9970 | 100 | 0.9972 |
| RL | 71.05 | 86.04 | 0.7786 | 99.98 | 0.9946 | 100 | 0.9956 |
| IU | 71.89 | 99.20 | 0.9702 | 100 | 0.9894 | 100 | 0.9912 |
| $l_1$-sparse | 69.60 | 90.10 | 0.7404 | 99.98 | 0.9704 | 99.98 | 0.9756 |
| SalUn | 71.99 | 88.72 | 0.7936 | 99.94 | 0.9890 | 100 | 0.9914 |

| Tiny-ImageNet | No Atk | | | 8/255 | | 16/255 | |
|---|---|---|---|---|---|---|---|
| | TA | UA | MIA | UA | MIA | UA | MIA |
| Original | 64.17 | 99.98 | 0.9978 | - | - | - | - |
| retrain | 61.81 | 60.17 | 0.6387 | 99.97 | 0.9735 | 100 | 0.9811 |
| FT | 55.66 | 85.42 | 0.8908 | 99.99 | 0.9969 | 99.97 | 0.9969 |
| RL | 55.36 | 72.88 | 0.8002 | 99.99 | 0.9962 | 99.98 | 0.9968 |
| IU | 56.33 | 94.85 | 0.9591 | 99.97 | 0.9967 | 99.98 | 0.9969 |
| $l_1$-sparse | 56.04 | 61.71 | 0.3836 | 99.99 | 0.7597 | 100 | 0.7614 |
| SalUn | 54.94 | 66.99 | 0.6237 | 99.99 | 0.9654 | 99.99 | 0.9681 |

Table 2: The Test Accuracy (TA), Unlearning Accuracy (UA), and MIA score of different baselines before and after Unlearning Mapping Attack for the Instance Unlearning scenario. The attack is bounded with a maximum strength of 8/255 and 16/255. The original here indicates the model performance before unlearning.

Accuracy we used here is the model's pure accuracy on the forget dataset and should be considered the lower the better.

On the other hand, for **generative tasks**, we focus on image generation. We adopt class-wise unlearning using Masked AutoEncoder(MAE) on ImageNet1k dataset. We use I2I (Li et al., 2024) and SalUn (Fan et al., 2024) as our image generation unlearning baselines. Empirical comparisons and numerical evaluations including inception score (IS) (Salimans et al., 2016) and Fréchet inception distance (FID) (Heusel et al., 2017) metrics are used to evaluate image generation quality. For performance comparison, IIS is higher the better; while lower FID indicates good generation quality. Please refer to the Appendix A.3 for more information.

## 5.2 EXPERIMENT RESULT

As shown in Table 1, class-level unlearning for discriminative tasks generally shows strong robustness against unlearning mapping attacks, especially in the case of retraining methods. This not only provides empirical evidence that a model can satisfy the robust unlearning criteria but also validates that class-wise discriminative unlearning effectively aligns the unlearned model with a retrained model. Note that we also include Test Accuracy(TA) in our evaluation metrics. While TA does not indicate any robustness of an unlearning method, it does reflect the strength of the unlearning. Since most unlearning methods have some hyperparameters that can adjust their strength, with stronger unlearning resulting in lower test accuracy, one can perform very strong unlearning to have good unlearning performance (low UA and MIA) and very low test accuracy which is impractical. Therefore, TA serves as a balancing metric, where lower TA can imply stronger unlearning, but the trade-off needs to be considered for robustness.

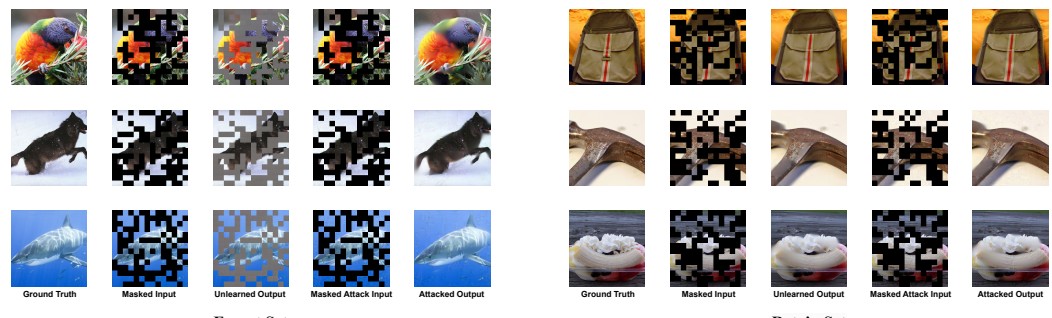

Figure 2: Unlearning Mapping Attack on image generation unlearning. I2I (Li et al., 2024) unlearning method is tested here. Reconstructed images are from ImageNet1k dataset.

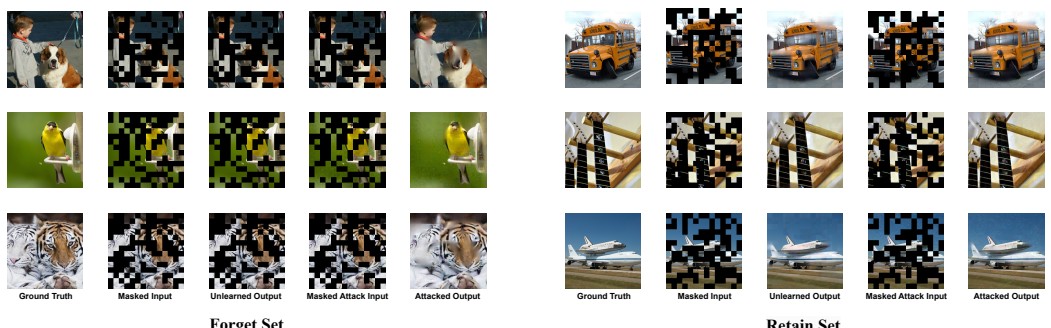

Figure 3: Unlearning Mapping Attack on image generation unlearning. SalUn (Fan et al., 2024) unlearning method is tested here. Reconstructed images are from ImageNet1k dataset.

For instance-level unlearning, as shown in Table 2, all baseline methods display limited robustness against unlearning mapping attacks. While the retraining method performs the best, it still lacks sufficient robustness, even with $\epsilon = 8/255$. This suggests that attackers can easily manipulate unlearned images, causing the model to re-recognize them, thus compromising the unlearning process.

Figure 2, 3 and Table 3 demonstrate our attack results on the image generation unlearning methods. Even when the unlearned model successfully avoids reconstructing images from the forget set, it can still generate well-restored images when fed an attacked input. Note that the attack can be unbounded for generation tasks since we only care about the model's output. However, as shown in both figures, only a small amount of noise is needed for the attack to succeed, highlighting the model's vulnerability to such attacks. Furthermore, the unlearning mapping attack does not significantly affect samples from the retain set, meaning there is no need to distinguish between the forget set and the retain set for this attack to be effective.

## 5.3 EMPIRICAL STUDY TOWARDS ROBUST UNLEARNING

Previous experiments reveal the possible hidden threat in current unlearning methods. In this section, we present our preliminary efforts, in terms of adversarial training and test-time sample pontification, to enhance the robustness of unlearning systems. The former targets to directly improve the robustness of unlearning methods by altering the optimization trace, and the latter emphasizes guarding a vulnerable unlearned model via a denoising module.

**Adversarial unlearning.** Intuitively, this approach builds on adversarial training (Madry et al., 2017), optimizing the original unlearning objective and the robust unlearning term simultaneously:

$$\min_{f_u} \mathbb{E}\{L_u(f_u(x_u), y) + \max_{\delta \in \mathbb{A}(f, f_u, x)} L_u(f_u(\delta), y)\}, \tag{5}$$

| | IS | | | | | | FID | | | | | |
|---|---|---|---|---|---|---|---|---|---|---|---|---|
| | No Atk | | 8/255 | | Unbound | | No Atk | | 8/255 | | Unbound | |
| | R | F | R | F | R | F | R | F | R | F | R | F |
| Original | 6.21 | 6.39 | - | - | - | - | 96.12 | 103.38 | - | - | - | - |
| I2I | 6.18 | 2.79 | 6.21 | 6.22 | 6.20 | 6.35 | 100.15 | 306.43 | 94.76 | 114.16 | 96.98 | 110.29 |
| SalUn | 6.05 | 2.42 | 6.02 | 6.11 | 6.13 | 6.27 | 130.45 | 330.79 | 102.75 | 133.82 | 94.15 | 108.44 |

Table 3: IS and FID results for Unlearning Mapping Attack on image generation unlearning. *R* and *F* stand for retain set and forget set. The attack strength is set to 0, 8/255, and unbound where the noise strength is unlimited. Note that a higher IS score indicates better image quality, while a lower FID score reflects improved image fidelity

| | No Atk | | | 8/255 | | 16/255 | |
|---|---|---|---|---|---|---|---|
| | TA | UA | MIA | UA | MIA | UA | MIA |
| RL-cifar10 | 91.53 | 0 | 0 | 1.44 | 0.0032 | 16.98 | 0.0532 |
| RL-cifar100 | 71.15 | 0 | 0.0004 | 20.28 | 0.1088 | 27.32 | 0.2348 |

Table 4: Empirical experiment on Adversarial Unlearning implemented in (5). RL(Golatkar et al., 2020) serves as a backbone unlearning system and is tested on both CIFAR10 and CIFAR100 datasets. Compared to the results in Table 1, the robustness of the system increases substantially.

where $L_u$ is the unlearning loss term adopted from an existing unlearning method, $x_u$ are the samples to be unlearned, and $\delta$ is the malicious attack input generated using $\mathbb{A}$, the attack algorithm described in Section 4.3. Our preliminary results, presented in Table 4, demonstrate a significant improvement in system robustness when using robust unlearning. We use the unlearning method from Golatkar et al. (2020) as our backbone with the addition of the robust unlearning term. Compared to the original results in Table 1, the robustness of the system increases substantially under the robust unlearning framework.

Adversarial unlearning can generally be applied to most existing unlearning methods, provided that the forget set is used during the unlearning process. Therefore, methods such as retraining or fine-tuning, which do not directly use the forget set, cannot directly incorporate robust unlearning. It should be noted that the proposed adversarial unlearning incurs computational overheads due to the iterative adversarial sample generation in the unlearning process. The total overhead largely depends on the size of the forget set. For large-scale systems with extensive forget sets, the scalability of adversarial unlearning may be constrained by computational resources.

**Test-time sample purification.** Instead of modifying unlearning algorithms as adversarial unlearning, on-the-fly sample-based purification during inference helps counter UMA without touching the unlearning process. As UMA operates by crafting adversarial noise added to query samples during inference, applying UMA-targeted purifiers to all queries before feeding them to the unlearned model might remove this adversarial noise and prevent forgotten knowledge from resurfacing. Preliminary studies and their quantitative results in Appendix A.2 show the potential of data purification to mitigate the UMA threat. Due to its computational-friendly, test-time purification is a practical alternative toward robust unlearning systems.

## 6 CONCLUSION

In this paper, we uncover a critical vulnerability present in most Machine Unlearning methods. We introduce the Unlearning Mapping Attack, which is designed to extract unlearned information from a system following the unlearning process. Our experiments demonstrate that the Unlearning Mapping Attack can successfully retrieve information that was supposed to be forgotten, across both generative and discriminative tasks. Additionally, we conducted a preliminary study on plug-in robust unlearning, which enhances the system's resilience to such attacks. We hope our work draws attention to the importance of robustness in Machine Unlearning and encourages the integration of robustness considerations when designing and evaluating unlearning methods.

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
