## A APPENDIX

### A.1 ABLATION STUDY

Here, we perform ablation experiments on the two hyperparameters we use in UMA method, number of steps and step size. All experiments are done using discriminative models on CIFAR10 dataset. SalUn (Fan et al., 2024) is chosen as the baseline learning algorithm. All ablation experiments on step sizes have a fixed number of steps of 100, and all ablations on iteration numbers have a fixed step size of 1/255. Attack strength is set to 16/255 across all ablations.

As the results shown in Figure 4, the attack efficacy generally increases as the number of steps goes up. However, higher iteration numbers result in greater computation costs, which forms a trade-off that the attacker needs to make. On the other hand, as shown in Figure 5, the attack step size reaches its best performance, around 0.7/255 to 1/255. A larger step size will cause the attack to find an incorrect direction, reducing the attack efficacy, while a smaller step size will generally cause a slow convergence speed, requiring a larger iteration step to reach equivalent performance.

### A.2 EXTENSIVE STUDY ON MITIGATING UMA

The robust unlearning implementation in Section 5.3 can increase model robustness against Unlearning Mapping Attack while maintaining clean test accuracy. However, robust unlearning requires extra computation costs and may not scale well with large machine learning models. Here, we propose another baseline solution for mitigating UMA method. As UMA operates by crafting adversarial noise added to query samples during inference, applying UMA-targeted purifiers to all queries before they are passed to the unlearned model might remove this adversarial noise and prevent forgotten knowledge from resurfacing. Preliminary studies are done implementing an autoencoder-based purification method using a variational autoencoder(VAE). We verify the purification system on CIFAR-10 dataset using both FT (Warnecke et al., 2021) and SalUn (Fan et al., 2024) as our baseline unlearning methods.

Table 5 and 6 represent results of whether the attacker has full knowledge of the purification. Generally, the results show increases in robustness in both cases, with strong robustness when the attacker has no knowledge of the purification, though the system's test accuracy is slightly impacted due to the VAE's limited reconstruction ability.

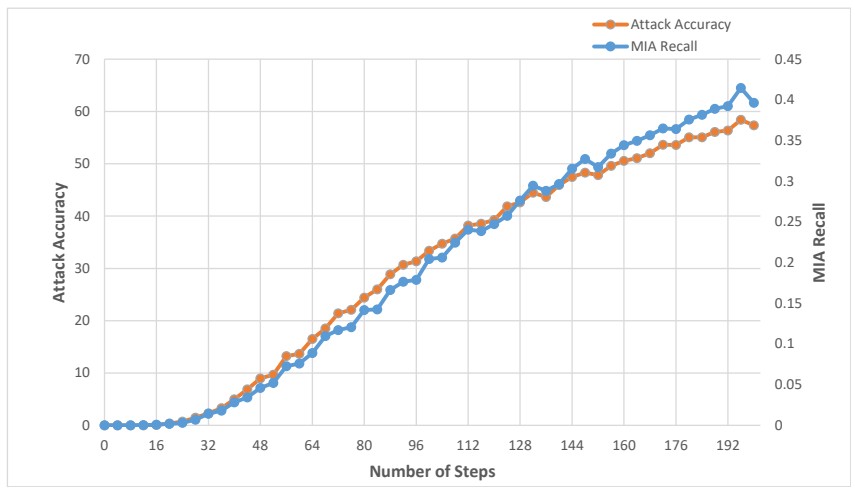

Figure 4: Ablation on attack iteration numbers. The experiments are done on CIFAR10 using SalUn (Fan et al., 2024) as the baseline unlearning algorithm. All experiments have a fixed step size of 1/255 and an attack strength of 16/255.

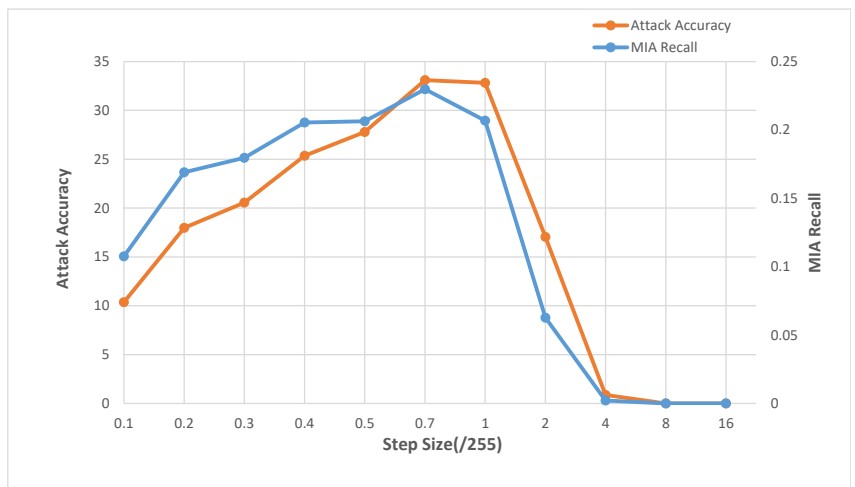

Figure 5: Ablation on attack step size. The experiments are done on CIFAR10 using SalUn (Fan et al., 2024) as the baseline unlearning algorithm. All experiments have a fixed number of steps of 100 and an attack strength of 16/255.

|  | No Atk | | 8/255 | | 16/255 | |
|---|---|---|---|---|---|---|
|  | UA | MIA | UA | MIA | UA | MIA |
| FT+vae | 4.86 | 0.0142 | 4.90 | 0.0142 | 5.40 | 0.0152 |
| SalUn+vae | 0 | 0 | 0 | 0 | 0 | 0 |

Table 5: Autoencoder-based UMA purification experiments on CIFAR-10. The attack has no knowledge of the purification.

|  | No Atk | | 8/255 | | 16/255 | |
|---|---|---|---|---|---|---|
|  | UA | MIA | UA | MIA | UA | MIA |
| FT+vae | 4.86 | 0.0142 | 99.98 | 0.9922 | 100 | 0.9970 |
| SalUn+vae | 0 | 0 | 0.66 | 0.0026 | 10.38 | 0.0584 |

Table 6: Autoencoder-based UMA purification experiments on CIFAR-10. The attack has full knowledge of the purification.

| L1 per image | ISI (Li et al., 2024) | | SalUn (Fan et al., 2024) | |
|---|---|---|---|---|
| | No Attack | 8/255 | No Attack | 8/255 |
| Retain set | 64,619 | 42,410 | 214,596 | 114,089 |
| Forget set | 1,140,778 | 48,317 | 2,790,552 | 242,029 |

Table 7: L1 norm between the outputs of the generative model before and after unlearning. The values under no attack are calculated by $L1(I_2, I_1)$ and the value under the attack strength 8/255 are computed by $L1(I_3, I_1)$.

### A.3 DETAILED INFORMATION AND RESULTS ON THE GENERATIVE UNLEARNING EXPERIMENTS

In the experiments on generative unlearning models, we evaluate if our UMA attacks could explore the residue information left in the model after unlearning and resurface the "forgotten" knowledge. To this end, we follow the previous arts in I2I where the generative model is used to recover the masked region in a query image. To ease the discussion, let's first clarify the data flow and pipeline of the generative model experiment. In our experiments, the generative unlearning pipeline involves the following steps:

- $I_0$: The ground truth image from the forget set.
- $I_m$: The masked version of the image $I_0$, which serves as the input to the generative model.
- $I_1$: The output of the original generative model (before unlearning), where the masked regions in $I_m$ are reconstructed.
- $I_2$: The output of the unlearned generative model, which cannot reconstruct the masked regions for the forget set and instead generates gray or noisy outputs.
- $I_3$: The output of the unlearned generative model when attacked with UMA, which aims to resurface the forgotten information and reconstruct the masked regions as $I_1$.

By design, $I_1$, $I_2$, and $I_3$ are naturally different from the masked input $I_m$, as the goal of the generative model is to reconstruct the missing regions. Additionally, for the forget set, $I_2$ differs significantly from $I_1$, as the unlearned model is intended to "forget" the knowledge and cannot recover $I_0$ from $I_m$. UMA's goal is to probe whether the unlearned model can generate $I_3$ that closely resembles $I_1$, thereby bypassing the unlearning mechanism. Based on the above context, UMA's efficacy is evaluated by how closely $I_3$ (the UMA output) resembles $I_1$ (the output of the original generative model before unlearning). This indicates whether the unlearned model retains residual knowledge of the forget set, effectively failing to fully "forget."

To verify UMA's impact, we directly computed the L1 distance between $I_3$ and $I_1$ per image. As shown in the Table 7, the L1 differences between $I_1$ and $I_3$ are very small after the attack (e.g. for the 224x224x3 image, average 0.3 intensity difference per pixel for the forget set with I2I (Li et al., 2024) and 1.6 intensity difference per pixel for the SalUn (Fan et al., 2024)), indicating that UMA can prompt the unlearned model to output information it was supposed to forget. This provides strong evidence that UMA effectively bypasses the unlearning process.

In addition, we include multiple visual examples in Figure 6 and 7. These examples present images for $I_0$, $I_m$, $I_1$, $I_2$, and $I_3$, providing a clear comparison of the reconstruction results across all stages of the pipeline. These visualizations demonstrate how UMA successfully recovers information that should have been forgotten, illustrating its effectiveness in attacking the unlearning mechanism.

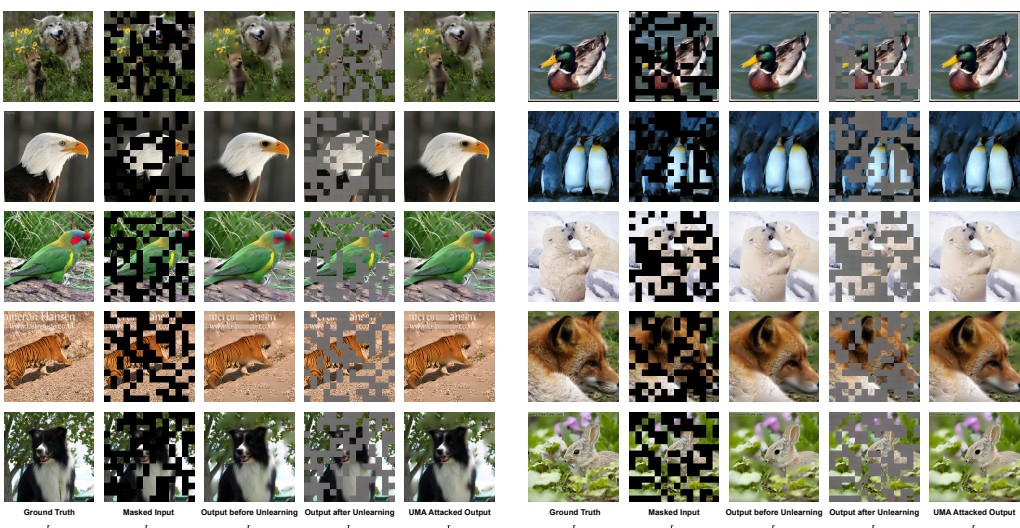

Figure 6: Examples of the generated images using I2I (Li et al., 2024) unlearning methods. Ground truth, $I_0$, Masked Input, $I_m$, Output before Unlearning, $I_1$, Output after Unlearning, $I_2$, UMA Attacked Output, $I_3$, are represented here as discussed in Section A.3

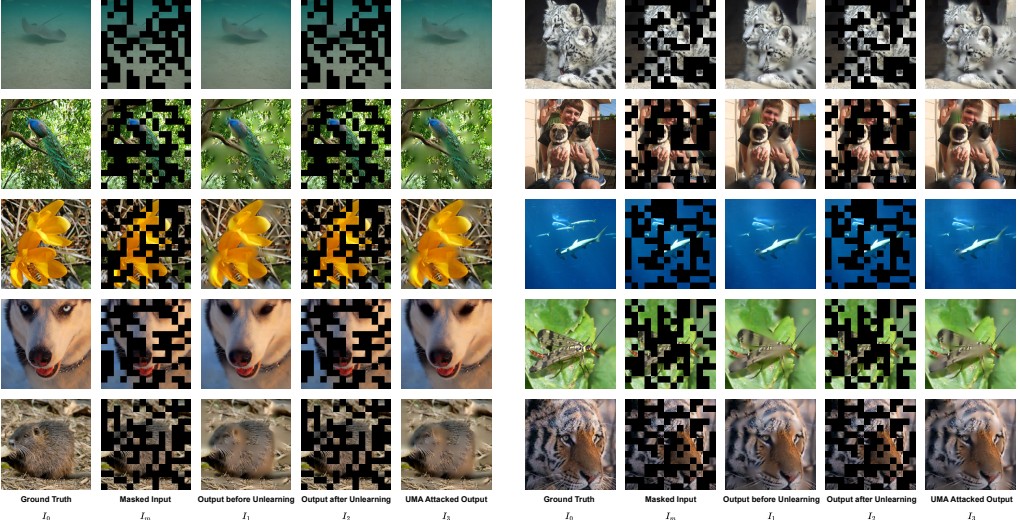

Figure 7: Examples of the generated images using SalUn (Fan et al., 2024) unlearning methods. Ground truth, $I_0$, Masked Input, $I_m$, Output before Unlearning, $I_1$, Output after Unlearning, $I_2$, UMA Attacked Output, $I_3$, are represented here as discussed in Section A.3