# OpenReview forum: "Unlearning Mapping Attack: Exposing Hidden Vulnerabilities in Machine Unlearning"
_ICLR.cc/2025/Conference — Submitted to ICLR 2025_

### Official Review · Reviewer_2sVh · 2024-10-25

**Soundness:** 2
**Presentation:** 3
**Contribution:** 3
**Rating:** 5
**Confidence:** 5

**Summary:**

This paper studies a new machine unlearning attack called unlearning mapping attacks. Specifically, an adversary given the full knowledge of the original and unlearning model, along with the unlearned data, can perturb the unlearned data as input to the unlearned model for reproducing the same output as the original model. The proposed attacks are achieved by adapting techniques from adversarial examples. Extensive experimental results validate the effectiveness of the proposed attacks. This paper studies a new machine unlearning attack called unlearning mapping attacks. Specifically, an adversary given the full knowledge of the original and unlearning model, along with the unlearned data, can perturb the unlearned data as input to the unlearned model for reproducing the same output as the original model. The proposed attacks are achieved by adapting techniques from adversarial examples. Extensive experimental results validate the effectiveness of the proposed attacks.

**Strengths:**

+ This paper proposes a new attack targeting machine unlearning. Specifically, an attacker can perturb the original unlearned data to make the unlearned model generate the same output as the data that has not been unlearned. This attack reveals the vulnerability of existing machine unlearning mechanisms.

+ The proposed attack is evaluated on several vision tasks across various datasets. The experimental results support the claims of the paper.

+ A defense mechanism is explored to mitigate the proposed attack and show somewhat effectiveness.

**Weaknesses:**

I have several concerns about this paper, which are listed below.

- Attack motivation. As listed in the introduction, the authors argue that "As a post-MUL attack, UMA only requires access to the model before and after the unlearning process, which practically makes the model provider or the MUL service provider an ideal candidate for carrying out such an attack." I do not understand why the server wants to launch the mapping attack. What is the benefit from the server's perspective?

- Threat model. The adversarial (i.e., the server) is given full access to the model before and after the unlearning process. In addition, specific samples that need to be unlearned are given to the attacker. In this case, for the attack goal, why did the server not directly plug backdoors into the unlearned model so a targeted attack could be achieved once the unlearned model is released? This could be an "active" attack, while under this assumption it seems more practical. In addition, as the authors describe, "here, we assume the attacker’s goal is to extract the forgotten information from the unlearned models," the attacker under this assumption already knows everything about the unlearned samples, why still need to perform such an attack?

- Experimental validation. In generative cases, is the model performing sample unlearning? Can the proposed attack work in retraining cases? In addition, the results are not that convincing. In Figure 2, the attacked output is more like removed masks of unlearned outputs, or inpainting from unlearned outputs.

**Questions:**

The questions are listed in the weakness part. I believe solving these concerns can help to improve the quality of this paper.

---

> ### Author Response · Authors · 2024-11-23
> **Response to Reviewer 2sVh (1/2)**
>
> We thank the reviewer for the detailed review and constructive feedback. We hope that our response to the specific weaknesses and questions in the following, as well as our revision highlighted in orange in the paper, addresses your concerns adequately.
>
> ---
>
> ***[W1] Attack motivation. As listed in the introduction, the authors argue that "As a post-MUL attack, UMA only requires access to the model before and after the unlearning process, which practically makes the model provider or the MUL service provider an ideal candidate for carrying out such an attack." I do not understand why the server wants to launch the mapping attack. What is the benefit from the server's perspective?***
>
> **[Response]** We thank the reviewer for this important question. Below, we outline two key benefits of UMA from the server’s perspective. First, from the role of adversarial servers, with increasing attention to privacy regulations such as GDPR and CCPA, servers face strict obligations to comply with unlearning requests. Directly storing or reusing data from the forget set poses significant legal and operational risks. UMA provides a covert alternative by enabling the adversarial server to dynamically reconstruct unlearned information from residual knowledge embedded in the unlearned model. This allows the server to exploit forgotten data while avoiding detection during audits or inspections. Second, from the role of responsible servers, UMA also allows a server to verify the effectiveness of the unlearning process. By probing residual traces of the forget set, the server can identify and quantify any vulnerabilities in the unlearned model. This assessment helps determine whether forgotten information is still exploitable, providing insights to improve the robustness of the unlearning process.
>
> ---
>
> ***[W2.1] Threat model. The adversarial (i.e., the server) is given full access to the model before and after the unlearning process. In addition, specific samples that need to be unlearned are given to the attacker. In this case, for the attack goal, why did the server not directly plug backdoors into the unlearned model so a targeted attack could be achieved once the unlearned model is released? This could be an "active" attack, while under this assumption it seems more practical.***
>
> **[Response]** We thank the reviewer for raising this important question. While directly implanting backdoors into the unlearned model could compromise the unlearning process, backdoor attacks require the adversary to actively modify the model by implanting specific triggers or behaviors. Such modifications are more likely to be detected during audits or tests of the unlearned model, increasing the risk of exposing the adversary’s malicious intent. For example, techniques like Proof-of-Unlearning (PoUL) [1] can effectively identify irregularities caused by adding malicious noise to training or unlearning data. In contrast, UMA operates as a **passive attack**, which does not alter the unlearned model. Instead, it leverages residual knowledge left behind after the unlearning process. This approach makes UMA significantly stealthier, as the unlearned model remains ostensibly compliant with unlearning requirements, thereby reducing the likelihood of detection.
>
> It is also important to note that while the server is the most likely adversary due to its privileged access to both the pre- and post-unlearning models as well as the forget set, the threat model is not strictly limited to the server. Anyone with access to these models could theoretically execute UMA. Thus, the UMA threat extends beyond server-side risk to encompass broader participant-side scenarios as well. However, executing a backdoor attack from such participant-side scenarios would be infeasible.
>
> Ref:
>
> *[1] Binchi Zhang, et al. Verification of machine unlearning is fragile. ICML, 2024.*
>
> ---
>
> ***[W2.2] In addition, as the authors describe, "here, we assume the attacker’s goal is to extract the forgotten information from the unlearned models," the attacker under this assumption already knows everything about the unlearned samples, so why still need to perform such an attack?***
>
> **[Response]** Please refer to our response to [W1] above. Thank you.

---

> ### Author Response · Authors · 2024-11-23
> **Response to Reviewer 2sVh (2/2)**
>
> ***[W3.1] Experimental validation. In generative cases, is the model performing sample unlearning? In addition, the results are not that convincing. In Figure 2, the attacked output is more like removed masks of unlearned outputs, or inpainting from unlearned outputs.***
>
> **[Response]** We are sorry for for any confusion or ambiguity in the generative experiments and Figure 2. For the generative experiments, we follow the unlearning experimental setup described in I2I [2] and conduct class-level unlearning. To verify the efficacy of UMA on unlearned models, we apply UMA directly on the image test set. That is, UMA directly modifies the input of the generative model, but not applied to the unlearned output.
>
> To clarify the data flow and pipeline of the generative model unlearning experiment, the experimental process can be summarized as follows:
> * $I_0$ : The ground truth image from the forget set.
> * $I_m$ : The masked version of $I_0$ , which serves as the input to the generative model.
> * $I_1$ : The output of the generative model (before unlearning), where the masked regions in $I_m$ are reconstructed and closely resembles $I_0$.
> *  $I_2$ : The output of the generative model (after unlearning), which fails to reconstruct the masked regions of $I_m$ in the forget set (e.g. producing gray or noisy outputs), but performs correctly for the retain set.
> * $I_3$ : The output of the generative model  (after unlearning) when the input $I_m$ is altered using UMA. In this case, $I_3$ can resurface the forgotten information in $I_m$, reconstructing the masked regions as $I_1$.
>
> By design, $I_2$ and $I_3$ are the outputs from the same unlearned model. The difference lies in their inputs, as UMA modifies $I_m$ for $I_3$. Specifically, for the forget set, $I_2$ is substantially different from $I_1$, as the unlearned model is intended to "forget" the knowledge and cannot reconstruct $I_0$ from $I_m$. UMA, however, edits the input $I_m$ to exploit residual knowledge left in the unlearned model, resulting in $I_3$ that closely resembles $I_1$ and bypasses the unlearning mechanism.
>
> In this context, UMA's effectiveness should be evaluated by comparing $I_3$ to $I_1$. A close resemblance between these outputs suggests that the unlearned model retains residual knowledge of the forget set, indicating a failure to completely "forget."
>
> To further clarify and address potential ambiguity, we have included detailed information on our generative model unlearning experiment and shown additional quantitative results and qualitative (visual) examples in the Appendix A.3.
>
> Ref:
>
> *[2] Guihong Li, et al. Machine unlearning for image-to-image generative models. ICLR, 2024.*
>
> ---
>
> ***[W3.2] Can the proposed attack work in retraining cases?***
>
> **[Response]** While UMA is primarily designed to probe unlearned models and resurface forgotten data in machine unlearning, it can also be applied to retrained models. In the case of retraining, UMA can verify whether the retrained model inadvertently retains the ability to generate knowledge not present in the retain set. In addition, comparing the behavior of a retrained model and an unlearned model on the same forget set provides valuable insights on robustness of the unlearning. This comparison helps determine whether the observed vulnerabilities stem from the inherent limitations of the unlearning algorithm or the characteristics of the dataset.

---

> > ### Author Response · Authors · 2024-12-02
> > **Additional Response to Reviewer 2sVh**
> >
> > Dear Reviewer 2sVh,
> >
> > We deeply appreciate your valuable insights and comments. With the reviewer response period set to conclude in approximately 28 hours, we would like to further utilize OpenReview’s interactive feature to engage in discussions with you, confirm if our response sufficiently addresses your concerns. Specifically, we have undertaken the following efforts to address your points:
> >
> > 1. Clarified the attack motivation from the server-side perspective.
> > 2. Clarified the threat model and the benefit of UMA as a passive attack (revised Section 4.1)
> > 3. Provided additional details on the experimental setup and validation in the generative model unlearning experiments (Appendix A.3).
> > 4. Discussed the potential application of UMA to the retraining set.
> >
> > We would greatly appreciate hearing back from you before the deadline and are happy to provide any additional clarifications if needed. Thank you.
> >
> > Best Regards,
> >
> > Authors of UMA

---

### Official Review · Reviewer_5VUm · 2024-10-29

**Soundness:** 2
**Presentation:** 3
**Contribution:** 2
**Rating:** 5
**Confidence:** 4

**Summary:**

This paper explores the vulnerability of machine unlearning (MUL) to malicious attacks that attempt to recover forgotten information. To address this question, this paper introduces the concept of robust unlearning. This paper also develops the unlearning mapping attack (UMA), a malicious attack against MUL that can retrieve forgotten information.

**Strengths:**

1.	This paper is clear and logically structured.
2.	This paper raises an important and noteworthy issue, namely whether MUL needs to be robust in both discriminative and generative tasks.

**Weaknesses:**

1.	The rationale behind the definition of robust unlearning requires further elaboration.
2.	The concept of robust unlearning does not seem to align well with the Unlearning Mapping Attack, as the strict conditions defined for robust unlearning are inconsistent with the optimization framework used in the attack.
3.	The Unlearning Mapping Attack method requires further explanation.
4.	It would be beneficial to include additional experiments with various generative models, such as DDPM that used by some baselines.

**Questions:**

1.	The paper mentions that a hidden weakness of MUL is that “successfully unlearned knowledge can resurface through appropriate probes.” However, if the retrain model also exhibits this limitation, then MUL is merely preserving this limitation. How do you ensure this issue is inherent to MUL itself and not to the model or data used?
2.	The concept of robust unlearning does not seem to align well with the Unlearning Mapping Attack. As I understand it, Propositions 2 and 3 require consideration of cases where an input might violate Condition 1. Thus, the definition of robust unlearning is aimed at ensuring that **any**   $x \in \mathcal{D}_u$ satisfies both Conditions 1 and 2. In contrast, the Unlearning Mapping Attack provides an optimization problem that seeks an input. If the solution obtained from this optimization is not optimal, it seems inconsistent with the definition of robust unlearning because there may still exist a solution that yields a lower loss. Consequently, the results of this attack do not align well with the definition of robust unlearning.
3.	I hope this paper can include additional explanation on how to apply the UMA method. Is this attack approach designed to generate a malicious dataset and then evaluate different MUL methods on this dataset?

---

> ### Author Response · Authors · 2024-11-23
> **Response to Reviewer 5VUm (1/2)**
>
> We thank the reviewer for the detailed review and constructive feedback. We hope that our response to the specific weaknesses and questions in the following, as well as our revision highlighted by magenta in the paper, addresses your concerns adequately.
>
> ---
>
> ***[W1] The rationale behind the definition of robust unlearning requires further elaboration.***
>
> **[Response]** Thanks for raising this question as it provides us an opportunity to elaborate on this matter. The standard definition of machine unlearning focuses on removing data in the forget set from a model, often equated with aligning the unlearned model’s behavior with that of a retrained model. However, this definition does not account for the adversarial exploitation of residual knowledge within the unlearned model for unlearned knowledge resurfaces. These vulnerabilities expose a critical gap in existing definitions, necessitating a more robust framework. The concept of robust unlearning introduces additional safeguards to ensure that the model cannot resurface unlearned information, even under adversarial conditions. This is formalized by adding constraints on the adversarial behavior of the unlearning model:
>
> * For generative tasks, it requires that no crafted input $δ$ can cause the unlearned model $f_u(\cdot;θ^u)$ to produce outputs resembling the original model’s behavior for data in the forget set $\mathcal{D}_u$.
> * For discriminative tasks, it requires that even slightly perturbed inputs $x+δ$ do not allow the unlearned model to revert to outputs consistent with $f(x,θ)$ for $D_u$.
>
> The definition of robust unlearning generalizes the standard unlearning constraints and incorporates robustness against adversarial threats. It aligns with real-world applications where models must guarantee security and compliance under diverse operational conditions, for example, protecting forgotten privacy data from adversarial recovery supports regulations like GDPR’s “right to be forgotten.”
>
> ---
>
> ***[W2 & Q2] The concept of robust unlearning does not seem to align well with the Unlearning Mapping Attack, as the strict conditions defined for robust unlearning are inconsistent with the optimization framework used in the attack. As I understand it, Propositions 2 and 3 require consideration of cases where an input might violate Condition 1. Thus, the definition of robust unlearning is aimed at ensuring that any satisfies both Conditions 1 and 2. In contrast, the Unlearning Mapping Attack provides an optimization problem that seeks input. If the solution obtained from this optimization is not optimal, it seems inconsistent with the definition of robust unlearning because there may still exist a solution that yields a lower loss. Consequently, the results of this attack do not align well with the definition of robust unlearning.***
>
> **[Response]** We thank the reviewer for highlighting the distinction between the theoretical requirements of robust unlearning and the practical framework of the Unlearning Mapping Attack (UMA). In our opinion, UMA and robust unlearning are complementary rather than inconsistent. Robust unlearning provides the idealized goal, while UMA offers a practical means to evaluate adherence to this goal by probing for specific vulnerabilities. Together, they form a framework for both defining and verifying unlearning robustness in a practical manner. Specifically, robust unlearning defines a theoretical ideal by requiring resilience to all possible adversarial perturbations. In practice, however, verifying this universal condition is computationally intractable due to the complexity of exploring the entire input space. UMA addresses this by formulating an optimization-based algorithm to find specific adversarial inputs that challenge the unlearning process. While UMA does not guarantee that it finds the worst-case perturbation, it serves as a lower-bound verification tool: if the attack succeeds, it demonstrates a violation of the robust unlearning definition. Even in cases where UMA does not succeed, the absence of successful attacks strengthens the empirical evidence that the model may satisfy the robust unlearning criteria. However, it does not provide a formal guarantee.
>
> To respond to this concern, we added a clarification at the end of Section 4 about UMA as follows: “While UMA does not guarantee exploration of all possible perturbations, it provides a practical and actionable framework for identifying vulnerabilities in unlearning methods. Even in cases where UMA does not succeed, the absence of successful attacks strengthens the empirical evidence that the model may satisfy the robust unlearning criteria.”

---

> ### Author Response · Authors · 2024-11-23
> **Response to Reviewer 5VUm (2/2)**
>
> ***[W3 & Q3] The Unlearning Mapping Attack method requires further explanation. I hope this paper can include additional explanation on how to apply the UMA method. Is this attack approach designed to generate a malicious dataset and then evaluate different MUL methods on this dataset?***
>
> **[Response]** We appreciate the reviewer’s thoughtful question. UMA serves two primary purposes. First, it is designed to generate malicious noise that can exploit residual vulnerabilities in unlearned models, enabling the resurfacing of forgotten knowledge. An attacker can target one or more samples in the forget set ($\mathcal{D}_u$) and use UMA to craft corresponding adversarial perturbations for malicious purposes. Second, UMA functions as a practical and actionable verification tool for benchmarking the effectiveness of different MUL methods. By applying UMA to the same set of forget data, it can quantitatively assess and compare the vulnerabilities of various unlearning methods against adversarial probs. While UMA can be used to generate malicious datasets based on the same forget set $\mathcal{D}_u$, it is important to note that these datasets are model-specific and not universal.
>
> To respond to this comment, We added the above discussion about the adversarial and evaluative functionalities of UMA at the beginning of Section 4.
>
> ---
>
> ***[Q1] The paper mentions that a hidden weakness of MUL is that “successfully unlearned knowledge can resurface through appropriate probes.” However, if the retrain model also exhibits this limitation, then MUL is merely preserving this limitation. How do you ensure this issue is inherent to MUL itself and not to the model or data used?***
>
> **[Response]** We thank the reviewer for raising this insightful question. As we discussed at the end of Section 3.2, It is important to note, however, that the commonly used retraining method, a baseline in many existing unlearning techniques, does not inherently satisfy the criteria for Robust Unlearning. Due to the generalization capabilities of neural networks, even models trained without specific data may still generate corresponding information. To distinguish MUL-specific vulnerabilities from broader limitations inherent to the model or dataset, we can evaluate both the unlearned model and a retrained model (trained solely on the retained set $D_r$) under UMA. This comparative analysis would suggest if similar vulnerabilities appeared in the retrained model, indicating that they are inherent to the model architecture and retained data or an incomplete unlearning process. For example, according to our empirical experiments on discriminative models with class-wise unlearning in Table 1, retraining achieves extraordinary robustness, but the evaluated machine unlearning algorithms suffer from UMA. This suggests that vulnerability is inherent to MUL.
>
> To respond to this comment, we extended our discussion at the end of Section 3.

---

> > ### Comment · Reviewer_5VUm · 2024-11-25
> > **Thanks for your responses**
> >
> > Thank you for your detailed response to my review and the improvements made to the manuscript. However, I still have some concerns regarding **W2** and **Q2**.
> >
> > As you mentioned, whether malicious perturbations against the model can be identified or not, we still cannot definitively conclude that the unlearning model is robust. Empirical results can only highlight which models are not robust but do not fully establish robustness.
> >
> > I believe this indicates a gap between the definition of "robust unlearning" and the concept of UMA. To address this, it would greatly strengthen the paper to provide a theoretical guarantee specifying the conditions under which a model can be considered "robust."

---

> ### Author Response · Authors · 2024-11-25
> **Response to Rebuttal Feedback**
>
> We would like to thank the reviewer for the insightful feedback and for emphasizing the importance of bridging the gap between the definition of "robust unlearning" and the practical implementation of gradient-based UMA attach algorithm. The reviewer’s comment that our gradient-based UMA algorithm cannot fully establish robustness in general is correct. And we further address this concern in two parts.
>
> (1) UMA and Robust Unlearning are consistent in principle. The UMA formulation in Equation (2) aligns with the definition of robust unlearning. Without loss of generality, let’s take the generative model as an example (we can get the same conclusion for the discriminative model). Following the definition, Robust unlearning requires that for each sample $x\in\mathcal{D}_u$ and all perturbations $δ_x$: $||f_u (δ_x ,θ^u)−f(x,θ)|| >ε_1 $. The UMA optimization problem seeks the perturbation $δ_x$  that minimizes: $||f_u (x+δ_x ,θ_u)−f(x,θ)||$. Suppose for every sample $x$ in the forget set, we find an optimal $δ_x$  that minimizes this difference, and the minimum difference is still larger than $ε_1$, we can conclude that the unlearned model satisfies the definition of robust unlearning with respect to $ε_1$ (namely $ε_1$-robustness). This consistency indicates that, under ideal conditions (e.g., if the optimization objective is convex), UMA provides a theoretical guarantee of robustness (with respect to $ε_1$). Note, that a practical setting of $ε_1$ should ensure that the robustness evaluation aligns with real-world attack detectability and human interpretability (e.g. MIA attacks or human visual inspection).
>
> (2) In practice, however, the optimization problem is typically non-convex due to the complexity of modern machine learning models. In such cases: (a) It becomes computationally intractable to search all possible perturbations $δ_x$ exhaustively, and (b) the gradient-based UMA implementation proposed in the paper may only find a local minimum rather than a global optimum. As a result, even if UMA identifies a perturbation $δ_x$  for which the difference is greater than $ε_1$ , we cannot definitively conclude $ε_1$-robustness because other, undetected perturbations might violate the condition. This is also the reason that in our experimentation section, our results generally focus on testing the attack efficacy instead of proving model robustness.
>
> To summarize:
> * The formulation of ideal UMA in equation (2) and the definition of robust unlearning are consistent. If the optimization problem were convex and we could exhaustively explore all perturbations, the ideal UMA attack would provide a theoretical guarantee for $ε_1$-robustness.
> * While our gradient-based UMA implementation is a powerful tool for identifying vulnerabilities and providing empirical evidence of robustness, its reliance on practical optimization techniques prevents it from offering formal, exhaustive guarantees.
>
> To respond to this comment, we will revise the manuscript in Section 4.2 & 4.3 (highlighted by magenta) to more clearly highlight these points, including the condition for the theoretical guarantees of robust unlearning and the practical limitations of UMA as an empirical verification tool.

---

> > ### Author Response · Authors · 2024-12-02
> > **Additional Response to Reviewer 5VUm**
> >
> > Dear Reviewer 5VUm,
> >
> > We deeply appreciate your valuable insights and comments. With the reviewer response period set to conclude in approximately 28 hours, we would like to further utilize OpenReview’s interactive feature to engage in discussions with you, confirm if our response sufficiently addresses your concerns. Specifically, we have undertaken the following efforts to address your comments:
> >
> > 1. Clarified the reasonable behind Robust Unlearning
> > 2. Explained the alignment between Robust Unlearning and UMA, as well as the implementation limitations in the  gradient-based UMA (revised Sections 4.2 and 4.3 on page 6).
> > 3. Provided clarification on the practical usage of UMA (revised Section 4 on page 5).
> > 4. Discussed potential sources of vulnerability and methods for identifying them (revised Section 3.3 on page 5)
> >
> > We would greatly appreciate hearing back from you before the deadline and are happy to provide any additional clarifications if needed. Thank you.
> >
> > Best Regards,
> >
> >  Authors of UMA

---

### Official Review · Reviewer_8EJ6 · 2024-11-03

**Soundness:** 3
**Presentation:** 3
**Contribution:** 3
**Rating:** 6
**Confidence:** 3

**Summary:**

This paper proposes the Unlearning Mapping Attack (UMA), a novel post-unlearning adversarial attack designed to expose vulnerabilities in existing machine unlearning (MUL) methods. The authors demonstrate that UMA can undermine current MUL techniques without altering the unlearning procedure itself, reintroducing forgotten information. The study highlights the insufficiencies of existing unlearning strategies in safeguarding models against such attacks, especially when adversaries have access to both pre- and post-unlearning models. Experiments on both generative and discriminative tasks reveal how UMA can retrieve forgotten data, emphasizing the need for a more robust definition of machine unlearning.

**Strengths:**

1. Novel Attack Strategy: The introduction of UMA fills a gap in machine unlearning research by focusing on post-unlearning attacks that expose hidden vulnerabilities. Unlike previous work that manipulates the unlearning process or data, UMA operates post-hoc, revealing a new class of threats.

2. Comprehensive Evaluation: The experiments span multiple datasets (CIFAR-10, CIFAR-100, Tiny-ImageNet, and ImageNet-1k) and tasks (discriminative and generative). This comprehensive evaluation strengthens the generality of the findings.

3. Focus on Robustness: The authors highlight the need for a new standard of Robust Unlearning, which ensures that models remain secure against adversarial inputs post-unlearning. This is a valuable contribution to both academia and industry, as it pushes for higher standards of security in privacy-preserving ML.

**Weaknesses:**

1. Limited Practical Mitigation Discussion: While the paper demonstrates the UMA vulnerability well, the section on mitigating this attack could be further expanded. Though adversarial training is explored, the feasibility of integrating robust unlearning techniques into various MUL methods needs more discussion, especially for large-scale systems.

2. Assumptions on Attacker's Knowledge: The UMA attack assumes that the adversary has full knowledge of both the pre- and post-unlearning models, which may not always be realistic. Clarifying how the attack would function under more limited access (e.g., black-box settings) would enhance the paper’s practical relevance.

3. Focus on Specific Unlearning Methods: The study tests UMA primarily on select MUL methods like fine-tuning and retraining. It would be beneficial to see more experiments on emerging unlearning methods, especially in federated learning or distributed systems, where unlearning is more complex.

**Questions:**

1. How does the UMA attack perform in black-box scenarios where the adversary has only partial knowledge of the model pre- and post-unlearning? Could the attack still be successful with limited model access?

2. What are the computational overheads of incorporating robust unlearning into existing systems? How does this affect the scalability of machine unlearning in real-world applications?

---

> ### Author Response · Authors · 2024-11-23
> **Response to Reviewer 8EJ6 (1/2)**
>
> We thank the reviewer for the detailed review and constructive feedback. We hope that our response to the specific weaknesses and questions in the following, as well as our revision highlighted in red in the paper, addresses your concerns adequately.
>
> ---
>
> ***[W1 & Q2] Limited Practical Mitigation Discussion: While the paper demonstrates the UMA vulnerability well, the section on mitigating this attack could be further expanded. Though adversarial training is explored, the feasibility of integrating robust unlearning techniques into various MUL methods needs more discussion, especially for large-scale systems. What are the computational overheads of incorporating robust unlearning into existing systems? How does this affect the scalability of machine unlearning in real-world applications?***
>
> **[Response]** We thank the reviewer for their valuable comment. Our response addresses these concerns in two parts.
>
> (1) *Regarding Computational Overheads and Scalability of Robust Unlearning:* The proposed adversarial unlearning incurs computational overheads due to the iterative adversarial sample generation in the unlearning process. The total overhead largely depends on the size of the forget set. To quantify this, we conducted an experiment on the CIFAR-10 dataset with a forget set of 5000 samples. Incorporating adversarial unlearning, in this case, required 8.5 times the unlearning time and twice the memory compared to the standard unlearning method. These results indicate that, in large-scale systems with extensive forget sets, the scalability of robust unlearning may be constrained by computational resources.
>
> (2) *Regarding other computation-friendly and scalable remedies for UMA Risk:* To explore more cost-effective approaches, we further investigated on-the-fly noise purification during inference as a defense against UMA. As UMA operates by crafting adversarial noise added to query samples during inference, applying UMA-targeted purifiers to all queries before passing them to the unlearned model might remove this adversarial noise and prevent forgotten knowledge from resurfacing. In our preliminary studies, we implemented a VAE-based purification module [1] on the CIFAR-10 dataset to remove noise from forget training samples. As shown in Appendix A.2, test-time purification successfully reduces UMA's efficacy, though the system’s test accuracy is slightly impacted due to the VAE’s limited reconstruction ability.
>
> To respond to this comment, we have added a brief discussion on the scalability of adversarial unlearning and test-time purification in Section 5.3 and Appendix A.2.
>
> Ref:
>
> *[1] Liao, Fangzhou, et al. "Defense against adversarial attacks using high-level representation guided denoiser." CVPR, 2018.*
>
> ---
>
> ***[W2 & Q1] Assumptions on Attacker's Knowledge: The UMA attack assumes that the adversary has full knowledge of both the pre- and post-unlearning models, which may not always be realistic. Clarifying how the attack would function under more limited access (e.g., black-box settings) would enhance the paper’s practical relevance. How does the UMA attack perform in black-box scenarios where the adversary has only partial knowledge of the model pre- and post-unlearning? Could the attack still be successful with limited model access?***
>
> **[Response]** We thank the reviewer for this suggestion. The UMA method is specifically designed as a white-box attack that requires full knowledge of both the pre- and post-unlearning models. Its purpose is to expose vulnerabilities in current unlearning methods, as well as to serve as a validation tool for testing the robustness of a specific machine unlearning algorithm. We believe that this validation role makes UMA practically significant, as it provides actionable insights into the risks of incomplete unlearning and informs the development of more robust unlearning techniques. Since UMA is model-specific, it might not yield the desired results when used in black-box scenarios. However, we recognize the importance of exploring transferable attacks in black-box settings and will investigate whether such an approach is feasible in future work.

---

> ### Author Response · Authors · 2024-11-23
> **Response to Reviewer 8EJ6 (2/2)**
>
> ***[W3] Focus on Specific Unlearning Methods: The study tests UMA primarily on select MUL methods like fine-tuning and retraining. It would be beneficial to see more experiments on emerging unlearning methods, especially in federated learning or distributed systems, where unlearning is more complex.***
>
> **[Response]** We thank the reviewer for this insightful comment. Unlearning in federated or distributed systems introduces significant complexities, such as communication overhead and decentralized updates, which make it more challenging than centralized methods. Our current focus is on centralized methods like fine-tuning and retraining, as a foundational step to demonstrate UMA’s efficacy. We recognize the importance of extending UMA to distributed settings and will explore this in future work.

---

> > ### Author Response · Authors · 2024-12-02
> > **Additional Response to Reviewer 8EJ6**
> >
> > Dear Reviewer 8EJ6,
> >
> > We deeply appreciate your valuable insights and comments. With the reviewer response period set to conclude in approximately 28 hours, we would like to further utilize OpenReview’s interactive feature to engage in discussions with you, confirm if our response sufficiently addresses your concerns. Specifically, we have undertaken the following efforts to address your comments:
> >
> > 1. Expanded the practical mitigation strategies for UMA (Revised Section 5.3 on page 10)
> > 2. Clarified the motivation of white-box setting and possible black-box performance
> >
> > We would greatly appreciate hearing back from you before the deadline and are happy to provide any additional clarifications if needed. Thank you.
> >
> > Best Regards,
> >
> >  Authors of UMA

---

### Official Review · Reviewer_z2dQ · 2024-11-03

**Soundness:** 2
**Presentation:** 2
**Contribution:** 2
**Rating:** 3
**Confidence:** 4

**Summary:**

This paper introduces an attack strategy in the context of machine unlearning. The attack involves introducing a minimal perturbation to a sample that has been unlearned, during inference. This perturbation is designed to make the model's prediction on the perturbed, unlearned sample closely resemble the prediction it would have made prior to the unlearning process. The paper first presents several propositions regarding the attack, followed by an optimization step to obtain such malicious perturbations. Several experiments are conducted on both discriminative and generative models.

**Strengths:**

1. The paper is clear and easy to follow.

2. The paper utilizes several popular unlearning methods to demonstrate the efficacy of the proposed attack method.

**Weaknesses:**

1. This paper's contribution is somewhat limited. The type of attack that aims to recover unlearned information after the unlearning phase has been recently explored in both large language models (LLMs) and the field of generative models. This paper proposes such unlearning attacks on discriminative classification tasks and generative tasks. However, for discriminative classification tasks, the attack is relatively straightforward. Moreover, the paper does not discuss related work or provide comparisons with existing post-unlearning attacks on generative models, such as diffusion models.

2. The definition of unlearning requires clarification. The statement "a discriminative model must no longer recognize the unlearned samples" along with equations (1) and (2) seems to be weird. Most existing definitions of machine unlearning involve the retrained model (though there is some debate over this), which does not imply that the unlearned model must no longer recognize the unlearned samples.

3. The propositions presented in the paper are not formally stated. For instance, Proposition 2 suggests that a universal perturbation $\delta$ applies to all unlearned samples, which does not seem to be the case according to the paper.

4. Figure 1 is somewhat ambiguous. At first glance, it might be misunderstood that the perturbation is added before the training phase, rather than after the unlearning phase.

5. The paper lacks ablation studies, and none of the experiments report standard deviations.

**Questions:**

1. Please explain why the definitions of unlearning attacks for discriminative and generative models differ in terms of $\delta$ and $x+\delta$.

2. Black-box transfer experiments would be interesting to demonstrate the efficacy of the proposed attack.

3. Which Membership Inference Attack (MIA) method do you choose for the experiments?

4. In your experiments, do you employ data augmentation?

---

> ### Author Response · Authors · 2024-11-23
> **Response to Reviewer z2dQ (1/3)**
>
> We thank the reviewer for their detailed review and thought-provoking discussion. We hope that our response to the specific weaknesses and questions in the following, as well as our revision highlighted in blue in the paper, addresses your concerns adequately.
>
> ---
>
> ***[W1] This paper's contribution is somewhat limited. The type of attack that aims to recover unlearned information after the unlearning phase has been recently explored in both large language models (LLMs) and the field of generative models. This paper proposes such unlearning attacks on discriminative classification tasks and generative tasks. However, for discriminative classification tasks, the attack is relatively straightforward. Moreover, the paper does not discuss related work or provide comparisons with existing post-unlearning attacks on generative models, such as diffusion models.***
>
> **[Response]** Thank you for raising this point and providing us with the opportunity to clarify our contribution and its distinction from related work. As presented in our paper, we review various malicious attempts targeting failed machine unlearning published between 2021 and 2024. Unlike prior works, which often manipulate forgetting data prior to unlearning or modifying unlearning algorithms, our study focuses on post-unlearning attacks, aiming to expose hidden vulnerabilities in existing machine unlearning methods. To the best of our knowledge, there is only one concurrent work on a similar post-unlearning attack, titled "*Towards Robust Knowledge Unlearning: An Adversarial Framework for Assessing and Improving Unlearning Robustness in Large Language Models.*" This paper, published on arXiv in late August 2024, focuses on unlearning attacks specifically targeting large language models. We have discussed this work in both the Introduction and Related Work sections of our paper. Beyond this, we are unaware of any other studies proposing similar post-unlearning threats. If the reviewer is referring to other papers, we would greatly appreciate it if you could share their titles, as we strive to ensure a comprehensive review of the relevant literature.
>
> ---
>
> ***[W2] The definition of unlearning requires clarification. The statement "a discriminative model must no longer recognize the unlearned samples" along with equations (1) and (2) seems to be weird. Most existing definitions of machine unlearning involve the retrained model (though there is some debate over this), which does not imply that the unlearned model must no longer recognize the unlearned samples.***
>
> **[Response]** We appreciate the reviewer’s insightful observation and agree that retraining-based definitions are often considered the ideal benchmark for machine unlearning. However, as highlighted by prior works (e.g., Cao & Yang, 2015 [1]), retraining-based definitions ensure that the unlearned model aligns closely with a retrained model on the retain set but impose no explicit constraints on the forget set. Our phrasing, "no longer recognize the unlearned samples," was intended to emphasize the need for a significant decorrelation between the outputs of the unlearned model and the original model on the forget set. This serves as a practical and measurable criterion for evaluating the efficacy of forgetting, particularly in scenarios where retraining is computationally prohibitive or impractical. The information we would like to deliver along with the equations here is that empirically, the unlearned model should diverge sufficiently from its prior behavior on the forget set while preserving its performance on the retain set. These formulations build on the widely accepted retraining-based definition, introducing explicit constraints that provide a practical framework for measuring both forgetting and retention. To ensure greater clarity, we have revised the paper accordingly and reformulated Equations in Section 3.1 to reflect the above discussion.
>
> Ref:
>
> *[1] Yinzhi Cao and Junfeng Yang. Towards making systems forget with machine unlearning. In 2015 IEEE Symposium on Security and Privacy, pp. 463–480, 2015.*

---

> ### Author Response · Authors · 2024-11-23
> **Response to Reviewer z2dQ (2/3)**
>
> ***[W3] The propositions presented in the paper are not formally stated. For instance, Proposition 2 suggests that a universal perturbation applies to all unlearned samples, which does not seem to be the case according to the paper.***
>
> **[Response]** We appreciate the reviewer’s feedback and acknowledge that the phrasing of Proposition 2 & 3 may lead to some ambiguity. To clarify, the perturbation in Proposition 2 & 3 is sample-specific and not universal. Each sample $x\in \mathcal{D}_u$ can have a distinct perturbation $δ_x$ that satisfies the proposition's conditions. Specifically, the formulation in Proposition 2 should be:
> \begin{equation}
> \text{There may exist an input}\quad \delta_x \notin \mathcal{D}_u \quad \text{s.t.} \quad |f_u(\delta_x,\theta^u)-f(x,\theta)|_2<\varepsilon_1, \forall x\in \mathcal{D}_u
> \end{equation}
> Similarly, the proposition 3 should be:
> \begin{equation}
> \text{There may exist a small non-zero} \quad\delta_x \in \mathbb{R}^d  \quad (\text{i.e.} ||\delta_x|| < \epsilon)  \quad\text{s.t.} \quad|f_u(x+\delta_x,\theta^u)-f(x,\theta)|_2<\varepsilon_1, \forall x\in \mathcal{D}_u
> \end{equation}
>
> This makes it clear that $δ_x$ is not a single, universal perturbation applied to all samples in the forget set $D_u$, but rather a perturbation for each individual $x$ in the forget set. We revise the statement of Proposition 2 & 3 in the paper to remove this ambiguity and improve formal rigor.
>
> ---
>
> ***[W4] Figure 1 is somewhat ambiguous. At first glance, it might be misunderstood that the perturbation is added before the training phase, rather than after the unlearning phase.***
>
> **[Response]** We thank the reviewer for pointing out this potential source of ambiguity. To address this, we have revised Figure 1 in the paper to clarify the sequence of events and explicitly highlight that the perturbation (malicious noise) is added after the unlearning phase, during the testing stage.
>
> ---
>
> ***[W5] The paper lacks ablation studies.***
>
> **[Response]** In UMA, there are two hyper-parameters, the number of steps and step size, for crafting the adversarial perturbation. In Appendix A.1, we added two ablations to study their effects on UMA. Briefly, the attack efficacy generally increases as the number of steps goes up. However, higher iteration numbers result in greater computation costs, which forms a trade-off that the attacker needs to make. On the other hand, the attack step size reaches its best performance, around 0.7/255 to 1/255. A larger step size will cause the attack to find an incorrect direction, reducing the attack efficacy, while a smaller step size will generally cause a slow convergence speed, requiring a larger iteration step to reach equivalent performance.
>
> ---
>
> ***[Q1] Please explain why the definitions of unlearning attacks for discriminative and generative models differ in terms of $δ$ and $x+δ$.***
>
> **[Response]** We thank the reviewer for this thoughtful question. As discussed in Section 3.1, the difference lies primarily in the nature of the tasks performed by discriminative and generative models, as well as how the perturbation $δ$ affects their input-output relationships.
> * In discriminative models, the output (e.g., a classification label or object detection) depends directly on the features of the input $x$. For the attack to be meaningful, the perturbed input $x+δ$ must remain realistic and semantically valid within the context of the task. Without a constraint on $δ$, the perturbation could produce nonsensical inputs (e.g., an image that no longer resembles a valid object), making the attack results meaningless. The constraint ensures the attack tests whether the model has forgotten knowledge conveyed in $D_u$ while preserving the practical validity of the inputs.
> * On the other hand, Generative models produce outputs based on the input or prompt, with a primary focus on the generated output rather than the input itself. In this context, the attacker can use unconstrained $δ$ to craft arbitrary inputs that exploit residual knowledge in the unlearned model. The focus of the attack is solely on whether the model generates outputs resembling unlearned information, regardless of the realism of the crafted inputs.
>
> In the revision, we have added a brief discussion at the end of Section 3.1 to clarify this distinction:
> *"The constraint on $δ$ for discriminative models ensures that the attack remains meaningful and realistic, aligning with the practical use cases of these models. By contrast, generative models allow for unconstrained $δ$, as the attack focuses exclusively on the outputs generated from crafted inputs, irrespective of input realism."*

---

> ### Author Response · Authors · 2024-11-23
> **Response to Reviewer z2dQ (3/3)**
>
> ***[Q2] Black-box transfer experiments would be interesting to demonstrate the efficacy of the proposed attack.***
>
> **[Response]** We thank the reviewer for this suggestion. The UMA method is specifically designed as a white-box attack that requires full knowledge of both the pre- and post-unlearning models. Its purpose is to expose vulnerabilities in current unlearning methods or serve as a validation tool for testing the robustness of a specific machine unlearning algorithm. Since UMA is model-specific, it might not yield the desired results when used in black-box scenarios. We recognize the importance of exploring transferable attacks in black-box settings and will investigate whether such an approach is feasible in future work.
>
> ---
>
> ***[Q3] Which Membership Inference Attack (MIA) method do you choose for the experiments?***
>
> **[Response]** In our experiments, we implement a shadow-model-based MIA attack following the methodology outlined in the original MIA paper [1]. Specifically, we construct 20 shadow models, each sharing the same structure as the main model, and train them using 8000 randomly selected training samples for 20 epochs. To create the shadow dataset, we feed training data (representing seen samples) and test data (representing unseen samples) into the shadow models. The shadow dataset is composed of logits (the model’s raw output) and corresponding labels indicating whether a logit originates from a seen or unseen sample. Using this shadow dataset, we train evaluation models, one per class. Each evaluation model is a linear binary classifier that takes logits as input and predicts whether they originate from data used to train the model. Once trained, these MIA models are used to evaluate the actual model's outputs, and we record the recall percentage as the final MIA result.
>
> Ref:
>
> *[1] Shokri, Reza, et al. "Membership inference attacks against machine learning models." 2017 IEEE symposium on security and privacy (SP). pp. 3-18, 2017.*
>
> ---
>
> ***[Q4] In your experiments, do you employ data augmentation?***
>
> **[Response]** During the training and unlearning stages, we follow the standard data augmentation strategies including random crop, random flip, and random rotation. During the UMA attack, since it is in the inference stage, we do not apply any data augmentation to the data we test.

---

> > ### Comment · Reviewer_z2dQ · 2024-11-25
> > **Thanks for your responses**
> >
> > Thank you for the clarifications. However, I still have several concerns as follows:
> >
> > > W1 Existing Literature on Post-Unlearning Attacks
> >
> > __Please refer to the following papers:__
> >
> > [1] To Generate or Not? Safety-Driven Unlearned Diffusion Models Are Still Easy To Generate Unsafe Images ... For Now
> >
> > [2] Ring-A-Bell! How Reliable are Concept Removal Methods for Diffusion Models?
> >
> > [3] Probing Unlearned Diffusion Models: A Transferable Adversarial Attack Perspective
> >
> > [4] Circumventing Concept Erasure Methods For Text-to-Image Generative Models
> >
> > > Q2 About Black-box transfer experiments
> >
> > Thus, this aspect highlights a weakness of the paper. Although this paper focuses on a white-box attack, there is potential for conducting transfer experiments over various unlearning methods, which could provide additional insights.
> >
> > > Q3 About MIA methods
> >
> > It would be highly beneficial to employ the LiRA [5,6] method to verify the effectiveness of unlearning processes.
> >
> > [5] Membership Inference Attacks From First Principles
> >
> > [6] Towards unbounded machine unlearning

---

> ### Author Response · Authors · 2024-11-29
> **Response to Rebuttal Feedback (1/2)**
>
> ***[W1]: Existing Literature on Post-Unlearning Attacks***
>
> ***Please refer to the following papers:***
>
> ***[1] To Generate or Not? Safety-Driven Unlearned Diffusion Models Are Still Easy To Generate Unsafe Images ... For Now***
>
> ***[2] Ring-A-Bell! How Reliable are Concept Removal Methods for Diffusion Models?***
>
> ***[3] Probing Unlearned Diffusion Models: A Transferable Adversarial Attack Perspective***
>
> ***[4] Circumventing Concept Erasure Methods For Text-to-Image Generative Models***
>
> **[Response]** We thank the reviewer for bringing these important studies to our attention. We will incorporate them into our discussion for comprehensive review. To clarify the contributions of our paper, we summarize the similarities and differences between these works and ours as follows.
> All of the four studies highlighted by the reviewer specifically target vulnerabilities in diffusion models (DMs). They explore adversarial text prompts [1,2] or transferable adversarial embeddings [3,4] to circumvent content erasure via machine unlearning. These attack methods rely heavily on the unique properties of diffusion mechanisms, making them nontrivial to extend and generalize beyond DM architectures. Furthermore, none of these works addresses potential remedies for the vulnerabilities they expose.
>
> In contrast, our work targets a more generic context, addressing vulnerabilities of machine unlearning across both discriminative and generative models. We revisit the machine unlearning (MUL) concept and identify a critical blindspot in the conventional MUL definition for the vulnerability. To bridge the gap, we introduce the practical and feasible metric for robust unlearning and propose a unified white-box adversarial attack formulation by quantifying differences between pre- and post-unlearning models. This general formulation encompasses specific cases proposed in those DM-focused studies. For instance, when substituting the diffusion classifier loss [1], the noise-level difference in DMs [2,3], or the KL divergence between diffusion trajectory distributions [4], into our framework for pre- and post-unlearning difference quantification, our UMA becomes equivalent to the white-box attacks proposed in these studies. Additionally, our work uniquely discusses potential mitigation strategies, a critical area that remains unexplored in the studies cited.
>
> In response to this comment, we have updated our literature review in the Introduction and Related Work Section and clarified their relationship to our works as below. These updates are highlighted in blue in the revised manuscript for easy review. “Recent research has explored this question by focusing on specific domains such as diffusion models (DMs) and large language models (LLMs). In the context of DMs, several approaches have been proposed to exploit vulnerabilities in content erasure (Zhang et al., 2025; Tsai et al., 2024; Han et al., 2024; Pham et al., 2023), yet none address potential remedies for these threats.”
>
> ---
>
> ***[Q2] About Black-box transfer experiments***
>
> ***Thus, this aspect highlights a weakness of the paper. Although this paper focuses on a white-box attack, there is potential for conducting transfer experiments over various unlearning methods, which could provide additional insights.***
>
> **[Response]** We thank the reviewer for raising this point. While we acknowledge the significant value of transferable attacks in black-box settings, we respectfully clarify that the focus on a white-box scenario in this study is not a weakness but a deliberate design choice aligned with our study's objectives. Specifically, the primary goal of our work is to expose an important blindspot in the conventional definition of machine unlearning. The white-box scenario allows for precise and targeted evaluation, enabling UMA’s adversarial attacks to be highly tailored to the target unlearned model. This approach maximizes the effectiveness of uncovering residual weaknesses, though it inherently limits the transferability of the attack to other models or black-box scenarios. Between the tradeoff of tailoring and generalizability, this study prioritizes precise evaluation within the white-box context to align with the objectives of this study.
>
> We also acknowledge that exploring transferable attacks in black-box settings is an important direction and plan to investigate this as part of our ongoing research efforts.

---

> ### Author Response · Authors · 2024-11-29
> **Response to Rebuttal Feedback (2/2)**
>
> ***[Q3] About MIA methods***
>
> ***It would be highly beneficial to employ the LiRA [5,6] method to verify the effectiveness of unlearning processes.***
>
> **[Response]** We thank the reviewer for the suggestion. Following your recommendation, we conducted additional experiments to assess unlearning efficacy using two different LiRA-based metrics. For both metrics, larger values correspond to better forget quality, signifying that the unlearned model retains less residual knowledge about the forgotten data. Due to time constraints, all experiments were conducted on the CIFAR10 dataset. The observations from these experiments align with the trends observed in our MIA results reported in the paper. Below, we summarize the experiments and findings:
>
> * Experiment 1: We used the official online LiRA procedure [5,6] to evaluate the unlearned models. Specifically, for each sample in the forget set, we computed the likelihood ratio. Since the references do not provide a specific method to aggregate these sample-based likelihood ratios into a single score, we followed standard performance evaluation practices and calculated the mean and standard deviation (std) of the likelihood ratios across all forget samples. The results, presented in the following table, show that for different unlearning methods, the mean likelihood ratios decrease to varying extents after UMA attacks. This indicates that UMA effectively exposes vulnerabilities in the unlearning process. Additionally, the large standard deviation suggests varying levels of difficulty among forget samples for machine unlearning.
>
> | | No Atk (mean ± std) | 8/255 (mean ± std) | 16/255 (mean ± std) |
> | --- | --- | --- | --- |
> | Original | 1.217 ± 1.109 | - | - |
> | Retrain |	3.163 ± 3.753 | 3.348 ± 4.013 | 3.315 ± 3.908 |
> | FT | 3.292 ± 3.926 | 1.052 ± 0.752 | 1.188 ± 1.078 |
> | RL | 3.375 ± 4.137 | 3.107 ± 3.453 | 3.040 ± 3.312 |
> | Wfisher | 3.178 ± 3.861 | 1.254 ± 3.090 | 1.297 ± 1.217 |
> | L1sparse | 3.380 ± 4.133 | 2.850 ± 2.940 | 2.694 ± 2.677 |
> | SalUn | 3.378 ± 4.134 | 3.089 ± 3.416 | 3.021 ± 3.264 |
>
> * Experiment 2: We applied another LiRA-based metric, introduced in the NeurIPS 2023 Machine Unlearning Challenge [7], to evaluate forget quality. Using the official competition evaluation procedure and code [8], we computed the forget quality scores and presented the results in the following table. The results show that after UMA attacks, the forget quality scores dropped to 0, indicating that UMA successfully leveraged residual information in the unlearned model to reconstruct knowledge from the forget set. Please note that this forget quality metric is not applicable to the retrain model by nature.
> | | No Atk | 8/255 | 16/255 |
> | --- | --- | --- | --- |
> | Original | 0 | - | - |
> | FT | 14.88 | 0 | 0 |
> | RL | 19.66 | 0 | 0 |
> | Wfisher | 41.14 | 0 | 0 |
> | L1sparse | 18.16 | 0 | 0 |
> | SalUn | 19.16 | 0 | 0 |
>
> We trust these results sufficiently address the reviewer’s concerns
>
>
> Ref:
>
> *[5] Membership Inference Attacks From First Principles*
>
> *[6] Towards unbounded machine unlearning*
>
> *[7] NeurIPS 2023 machine unlearning challenge, https://unlearning-challenge.github.io/*
>
> *[8] Official evaluation code for the NeurIPS 2023 Machine Unlearning competition, https://github.com/google-deepmind/unlearning_evaluation*

---

> > ### Author Response · Authors · 2024-12-02
> > **Additional Response to Reviewer z2dQ**
> >
> > Dear Reviewer z2dQ,
> >
> > We deeply appreciate your valuable insights and comments. With the reviewer response period set to conclude in approximately 28 hours, we would like to further utilize OpenReview’s interactive feature to engage in discussions with you, confirm if our response sufficiently addresses your concerns. Specifically, we have undertaken the following efforts to address your comments:
> >
> > 1. Clarified the scope of this study and its relationship to prior works (revised Section 1 & 2)
> > 2. Provided clarification on the definition of unlearning and its implicit blind spots (revised Section 3.1 on page 3)
> > 3. Clarified the sample-based nature of UMA perturbations (revised Section 3.1 on page 4)
> > 4. Updated Figure 1 to better illustrate the system overview (revision on page 4).
> > 5. Added ablation studies (Appendix A.1)
> > 6. Elaborated on the motivation for constraining perturbation in discriminative models (Revised Section 3.1 on Page 4).
> > 7. Conducted additional quantitative experiments on MIA, with a focus on LiRA-based methods.
> >
> > We would greatly appreciate hearing back from you before the deadline and are happy to provide any additional clarifications if needed. Thank you.
> >
> > Best Regards,
> >
> > Authors of UMA

---

### Official Review · Reviewer_dxPa · 2024-11-04

**Soundness:** 2
**Presentation:** 3
**Contribution:** 2
**Rating:** 5
**Confidence:** 3

**Summary:**

This paper presents a novel adversarial attack, the Unlearning Mapping Attack (UMA), which exposes vulnerabilities in current machine unlearning techniques. Machine unlearning aims to remove specific data from trained models to ensure privacy and compliance with regulations. However, UMA demonstrates that even when data is supposedly "forgotten," it can be reconstructed by leveraging information from the model before and after unlearning. The study evaluates UMA on both generative and discriminative tasks, showing that existing unlearning methods are vulnerable to such attacks without any alteration to the unlearning process itself. The authors propose a need for "Robust Unlearning," which strengthens models against adversarial reconstruction. Experimental results highlight UMA's effectiveness in recovering forgotten data, underscoring the importance of reassessing current unlearning techniques to prioritize resistance against sophisticated attacks.

**Strengths:**

The paper’s strengths lie in its introduction of the Unlearning Mapping Attack (UMA), a novel strategy that uncovers vulnerabilities in current machine unlearning methods without altering the unlearning process. Through comprehensive testing on both generative and discriminative tasks, UMA demonstrates broad applicability and highlights the critical need for “Robust Unlearning” standards to defend against adversarial attacks. By assuming realistic attack conditions, the study provides valuable insights into practical security implications, contributing to the advancement of secure and resilient unlearning techniques.

**Weaknesses:**

Weakness 1: This method does not assess the resistance of existing defenses against poisoning samples.
Weakness 2: The format discrepancy between the attack output and input may allow service providers to plausibly deny the relationship between them. Simple metrics, such as L1 distance, could easily indicate that the input and output are not the same.
Weakness 3: The relationship between the poisoning rate and the amount of data reconstructed remains unexplored.

**Questions:**

1 .Can sample-based defenses effectively mitigate the attack?
2. What is the relationship between the poisoning rate and data reconstruction after unlearning?
3. What is the L1 norm difference between the input and output?

---

> ### Author Response · Authors · 2024-11-23
> **Response to Reviewer dxPa (1/2)**
>
> We thank the reviewers for their constructive feedback. We will first clarify the scope and relevance of the Unlearning Mapping Attack (UMA) before responding to the specific weaknesses and questions. We hope that our response in the following along with our revision highlighted in red in the paper address your concerns adequately.
>
> **Scope of UMA:**
>
> UMA addresses a critical gap in machine unlearning research by exposing how incomplete unlearning can allow forgotten information to resurface, revealing vulnerabilities in current unlearning methods. Unlike data poisoning, UMA does not introduce malicious samples into the training data. Instead, it operates as a **post-unlearning attack**, crafting perturbations to probe residual vulnerabilities left by the unlearning process without altering the training or unlearning workflows. **This clearly differentiates UMA from poisoning-based attacks.** In addition to being an attack, UMA also serves as a **verification tool**, providing insights into the effectiveness of machine unlearning algorithms. This dual role makes UMA valuable both for identifying weaknesses and for guiding the development of more robust unlearning techniques.
>
> With these clarifications, we address the specific weaknesses and questions from the reviewer below.
>
> ---
>
> ***[W1 & Q1] This method does not assess the resistance of existing defenses against poisoning samples. Can sample-based defenses effectively mitigate the attack?***
>
> **[Response]** To the best of our knowledge, sample-based defenses against data poisoning attacks usually take place before the training process, such as data aggregation, data sanitization, and data augmentation. There are also defenses after training, but they generally relate to actions to the model like model validation. Since UMA is unrelated to poisoning attacks and occurs at the inference stage without altering the training/unlearning process, these sample-based defenses are not directly applicable to UMA.
>
> However, we acknowledge that general sample-based defenses against adversarial attacks such as data sanitization might help mitigate UMA. Specifically, we believe **on-the-fly sample-based purification during inference** can help counter UMA. As UMA operates by crafting adversarial noise added to query samples during inference, applying UMA-targeted purifiers to all queries before they are passed to the unlearned model might remove this adversarial noise and prevent forgotten knowledge from resurfacing. Our preliminary studies and quantitative results added in Appendix A.2 show the potential of data purification to mitigate the UMA threat.
>
> To respond to this comment, we added a brief discussion on test-time sample-based purification in Section 5.3 of the paper, which was highlighted in Red. The detailed experiments and results are presented in the appendix A.2.
>
> ---
>
> ***[W2 & Q3] The format discrepancy between the attack output and input may allow service providers to plausibly deny the relationship between them. Simple metrics, such as L1 distance, could easily indicate that the input and output are not the same. What is the L1 norm difference between the input and output?***
>
> **[Response]** The image generative experiments show that the unlearned model is incapable of recovering the forgetting data after model unlearning, but UMA probes the unlearned model and successfully prompts the model to recover the images. Though UMA does not aim to produce exact pixel-level reconstructions of forgotten data, we observe high **semantic similarity** between the input and their ground-truth images. In response to the reviewer’s comments, we report in the following table with the L1 norm difference per image between the attack outputs and their corresponding ground-truth images on two state-of-the-art generative unlearning methods published at ICLR’24: I2I [1] and SalUn [2].
>
> | L1-norm diff for I2I [1] | No Attack | 8/255 |
> | ---------------------------- | ------------- | ------- |
> | Forget set | 1248161 | 365104 |
> | Retain set | 316541 | 304539 |
>
> | L1-norm diff for SalUn [2] | No Attack | 8/255 |
> | ---------------------------- | ------------- | ------- |
> | Forget set | 2796692 | 495821 |
> | Retain set | 383969 | 327684 |
>
> The key observations are (1) After unlearning, both methods demonstrate large L1 norm values for the forget set, indicating that the models struggle to reconstruct forgotten classes. (2) When UMA is applied to the forget set (i.e., by adding adversarial noise to forgotten samples), the L1 norm values are significantly reduced to levels similar to the retain set. This indicates that UMA successfully leverages residual knowledge in the unlearned models to resurface forgotten samples.
>
> Ref:
>
> [1] Guihong Li et al. Machine unlearning for image-to-image generative models. ICLR, 2024.
>
> [2] Chongyu Fan, et al. Salun: Empowering machine unlearning via gradient-based weight saliency in both image classification and generation. ICLR, 2024.

---

> > ### Comment · Reviewer_dxPa · 2024-11-25
> > **Reviewer Feedback**
> >
> > I would like to raise a concern regarding the significance of generating a complete sample through reverse learning. For example, if the generated sample differs from the original input, it becomes unclear how meaningful the result is. Furthermore, a simple L1 distance comparison would allow the service provider to deny any connection to the provided input, arguing that they never generated such a sample. Without a clear way to attribute the generated sample back to the original input, it is difficult to establish its relevance. In this context, what is the practical significance or impact of this type of attack? Could you elaborate on this point?

---

> ### Author Response · Authors · 2024-11-23
> **Response to Reviewer dxPa (2/2)**
>
> ***[W3 & Q3] The relationship between the poisoning rate and the amount of data reconstructed remains unexplored. What is the relationship between the poisoning rate and data reconstruction after unlearning?***
>
> **[Response]** Since UMA is unrelated to data poisoning, the concept of a poisoning rate is not applicable to our work. UMA operates by crafting sample-specific perturbations for each data point in the forget set. As a result, there is no direct relationship between poisoning rates and UMA’s success.

---

> ### Author Response · Authors · 2024-11-26
> **Response to Feedback**
>
> Thank you for your thoughtful comment. To address your concern, we would like to clarify the evaluation pipeline and the relevance of UMA's output.
>
> To ease the discussion, let’s first clarify the data-flow and pipeline of the generative model experiment. In our experiments, the generative unlearning pipeline involves the following steps:
> * $I_0$ : The ground truth image from the forget set.
> * $I_m$ : The masked version of the image $I_0$ , which serves as the input to the generative model.
> * $I_1$ : The output of the original generative model (before unlearning), where the masked regions in $I_m$ are reconstructed.
> *  $I_2$ : The output of the unlearned generative model, which cannot reconstruct the masked regions for the forget set and instead generates gray or noisy outputs.
> * $I_3$ : The output of the unlearned generative model when attacked with UMA, which aims to resurface the forgotten information and reconstruct the masked regions as $I_1$.
>
> By design, $I_1$ , $I_2$ , and $I_3$ are naturally different from the masked input $I_m$, as the goal of the generative model is to reconstruct the missing regions. Additionally, for the forget set, $I_2$ differs significantly from $I_1$, as the unlearned model is intended to "forget" the knowledge and cannot recover $I_0$ from $I_m$. UMA's goal is to probe whether the unlearned model can generate $I_3$ that closely resembles $I_1$, thereby bypassing the unlearning mechanism.
>
> Based on the above context, UMA's success should not be judged by comparing $I_3$ to the original input of the generative model (i.e. the masked input $I_m$), as the masked input is inherently incomplete. Instead, UMA’s efficacy is evaluated by how closely $I_3$ (the UMA output) resembles $I_1$ (the output of the original generative model before unlearning). This indicates whether the unlearned model retains residual knowledge of the forget set, effectively failing to fully "forget."
>
> In our previous rebuttal, we calculated the L1 distance between $I_3$ and $I_0$ (ground truth) to assess whether the UMA attack produces outputs with similar reconstruction quality for the forget set and the retain set. The results showed that, after UMA attacks, the L1 differences between $I_3$ and $I_0$ (ground truth) for the forget set and retain set are at comparable levels, suggesting that the unlearned model retains residual knowledge of the forget set.
>
> To further verify UMA's impact, we directly computed the L1 distance between $I_3$  and $I_1$ per image. As shown in the table below, the L1 differences between $I_1$  and $I_3$ are very small after the attack, (for the 224x224x3 image, average 0.3 intensity difference per pixel for the forget set with I2I [1] and 1.6 intensity difference per pixel for the SalUn [2]), indicating that UMA can prompt the unlearned model to output information it was supposed to forget. This provides strong evidence that UMA effectively bypasses the unlearning process.
>
> | L1-norm diff for I2I [1] | No Attack ($L1(I_2,I_1)$) | 8/255 ($L1(I_3,I_1)$) |
> | ---------- | --------- | ---------- |
> | Forget set | 1140778 | 48317 |
> | Retain set | 64619 | 42410 |
>
> | L1-norm diff for SalUn [2] | No Attack ($L1(I_2,I_1)$) | 8/255 ($L1(I_3,I_1)$) |
> | ---------- | --------- | ---------- |
> | Forget set | 2790552 | 242029 |
> | Retain set | 214596 | 114089 |
>
> To further clarify and address potential ambiguity, we have included additional visual examples in the Appendix. These examples present images for $I_0$, $I_m$, $I_1$, $I_2$, and $I_3$, providing a clear comparison of the reconstruction results across all stages of the pipeline. These visualizations demonstrate how UMA successfully recovers information that should have been forgotten, illustrating its effectiveness in attacking the unlearning mechanism.
>
> We hope this explanation clarifies the significance of UMA in revealing vulnerabilities in generative unlearning. Thank you again for your comment, which allowed us to further refine our analysis.
>
> Ref:
>
> *[1] Guihong Li et al. Machine unlearning for image-to-image generative models. ICLR, 2024.*
>
> *[2] Chongyu Fan, et al. Salun: Empowering machine unlearning via gradient-based weight saliency in both image classification and generation. ICLR, 2024.*

---

> > ### Author Response · Authors · 2024-12-02
> > **Additional Response to Reviewer dxPa**
> >
> > Dear Reviewer dxPa,
> >
> > We deeply appreciate your valuable insights and comments. With the reviewer response period set to conclude in approximately 28 hours, we would like to further utilize OpenReview’s interactive feature to engage in discussions with you, confirm if our response sufficiently addresses your concerns. Specifically, we have undertaken the following efforts to address your comments:
> >
> > 1. Provided a detailed description of the relationship between UMA attack and data poisoning.
> > 2. Added a discussion on sample-based defense mechanisms against UMA (Revised Section 5.3).
> > 3. Clarified the experimental setup for generative model unlearning and evaluation (Appendix A.3).
> > 4. Conducted ablation studies quantifying the L1 norm in generative unlearning results (Appendix A.3).
> >
> > We would greatly appreciate hearing back from you before the deadline and are happy to provide any additional clarifications if needed. Thank you.
> >
> > Best Regards,
> >
> > Authors of UMA

---

### Meta-Review · Area_Chair_8fzz · 2024-12-12

**Metareview:**

This work studies an adversarial attack (i.e., UMA) against machine learning, highlighting vulnerabilities in the current unlearning process. Its strengths include the novel attack method and good writing. Nevertheless, the limited types of tasks it considers, the lack of ablation studies, and the strong assumption of adversarial knowledge indicate the weaknesses of this work. Overall, the reviewers generally agree that the current submission is not ready for publication.

**Additional Comments On Reviewer Discussion:**

Reviewers point out the limited evaluation models of this work (Reviewers z2dQ and 5VUm), the lack of effective mitigations and ablation experiments (Reviewers dxPa, z2dQ, and 8EJ6), and the adversary's reliance on white-box knowledge (Reviewers z2dQ, 8EJ6, and 2sVh). The authors provide additional explanations and experiments to partially address these weaknesses, but given its remaining weaknesses, this work needs to be improved to reach an acceptable level.

---

### Decision · Program_Chairs · 2025-01-22

Reject